# Antibodies inhibit transmission and aggregation of *C9orf72* poly-GA dipeptide repeat proteins

Qihui Zhou[1,2], Carina Lehmer[1], Meike Michaelsen[1], Kohji Mori[3,4], Dominik Alterauge[5], Dirk Baumjohann[5], Martin H Schludi[1,2], Johanna Greiling[1], Daniel Farny[1], Andrew Flatley[6], Regina Feederle[1,2,6], Stephanie May[1], Franziska Schreiber[1], Thomas Arzberger[1,7,8], Christoph Kuhm[1,2,9], Thomas Klopstock[1,2,9], Andreas Hermann[10], Christian Haass[1,2,3] & Dieter Edbauer[1,2,3,*]

## Abstract

Cell-to-cell transmission of protein aggregates is an emerging theme in neurodegenerative disease. Here, we analyze the dipeptide repeat (DPR) proteins that form neuronal inclusions in patients with hexanucleotide repeat expansion *C9orf72*, the most common known cause of amyotrophic lateral sclerosis (ALS) and frontotemporal lobar degeneration (FTLD). Sense and antisense transcripts of the $(G4C2)_n$ repeat are translated by repeat-associated non-ATG (RAN) translation in all reading frames into five aggregating DPR proteins. We show that the hydrophobic DPR proteins poly-GA, poly-GP, and poly-PA are transmitted between cells using co-culture assays and cell extracts. Moreover, uptake or expression of poly-GA induces nuclear RNA foci in $(G4C2)_{80}$-expressing cells and patient fibroblasts, suggesting an unexpected positive feedback loop. Exposure to recombinant poly-GA and cerebellar extracts of *C9orf72* patients increases repeat RNA levels and seeds aggregation of all DPR proteins in receiver cells expressing $(G4C2)_{80}$. Treatment with anti-GA antibodies inhibits intracellular poly-GA aggregation and blocks the seeding activity of *C9orf72* brain extracts. Poly-GA-directed immunotherapy may thus reduce DPR aggregation and disease progression in *C9orf72* ALS/FTD.

**Keywords** amyotrophic lateral sclerosis; C9orf72; immunotherapy; RAN translation; seeding

**Subject Category** Neuroscience

## Introduction

Intracellular protein aggregation is a common feature of Alzheimer's disease and many other neurodegenerative disorders. Cell-to-cell transmission of intracellular protein aggregates has been described for intracellular tau and α-synuclein aggregates forming amyloid fibrils (Chai *et al*, 2012; Sanders *et al*, 2014). The secretion and reuptake mechanisms are largely unknown, but the transmitted small aggregates seem to act as nucleation seeds that template further aggregation in the receiving cell (Jucker & Walker, 2011). The spreading of aggregates between cells is thought to cause the stereotypic progression of tau pathology through synaptically connected brain regions during disease progression (Braak *et al*, 2006; Iba *et al*, 2015; Takeda *et al*, 2015). Ongoing preclinical and clinical trials aim to interrupt the spreading of intraneuronal pathology using mostly passive vaccination (Yanamandra *et al*, 2013).

In 2011, a $(G4C2)_n$ repeat expansion upstream of the coding region of *C9orf72* was found to cause frontotemporal lobar degeneration (FTLD) and/or amyotrophic lateral sclerosis (ALS) in about 10% of all Caucasian patients with these related fatal neurodegenerative conditions (DeJesus-Hernandez *et al*, 2011; Renton *et al*, 2011; Gijselinck *et al*, 2012). *C9orf72* haploinsufficiency, toxic nuclear RNA foci, and translation into toxic dipeptide repeat (DPR) proteins have been suggested as drivers of pathogenesis (Edbauer & Haass, 2016). Animal models expressing the repeat expansion strongly support a gain-of-function mechanism (Mizielinska *et al*, 2014; Chew *et al*, 2015; Jiang *et al*, 2016; Liu *et al*, 2016). Repeat RNA accumulates in nuclear foci and sequesters several RNA-binding proteins (Mori *et al*, 2013b), but even high level expression of the repeat RNA from an intron is not toxic in *Drosophila* models

1 German Center for Neurodegenerative Diseases (DZNE), Munich, Germany
2 Munich Cluster of Systems Neurology (SyNergy), Munich, Germany
3 Biomedical Center, Biochemistry, Ludwig Maximilians-Universität München, Planegg-Martinsried, Germany
4 Department of Psychiatry, Osaka University Graduate School of Medicine, Osaka, Japan
5 Institute for Immunology, Biomedical Center Munich, Ludwig Maximilians-Universität München, Planegg-Martinsried, Germany
6 Monoclonal Antibody Core Facility and Research Group, Institute for Diabetes and Obesity, Helmholtz Zentrum München, German Research Center for Environmental Health (GmbH), Munich, Germany
7 Center for Neuropathology and Prion Research, Ludwig Maximilians-Universität München, Planegg-Martinsried, Germany
8 Department of Psychiatry and Psychotherapy, Ludwig Maximilians-Universität München, Planegg-Martinsried, Germany
9 Department of Neurology, Friedrich-Baur-Institute, Ludwig Maximilians-Universität München, Planegg-Martinsried, Germany
10 Department of Neurology and Center for Regenerative Therapies Dresden (CRTD), Technische Universität Dresden and German Center for Neurodegenerative Diseases (DZNE), Dresden, Germany
*Corresponding author. Tel: +49 89 440046510; E-mail: dieter.edbauer(at)dzne.de

(Tran *et al*, 2015). Sense and antisense repeat transcripts are translated in all reading frames into five aggregating DPR proteins (Ash *et al*, 2013; Mori *et al*, 2013a,c) by an unconventional mechanism. This so-called repeat-associated non-ATG (RAN) translation was first described for expanded CAG repeats and seems to require formation of RNA hairpins (Zu *et al*, 2011). Poly-GA is abundantly expressed in the *C9orf72* brains, followed by poly-GP and poly-GR, while poly-PA and poly-PR resulting from translation of the antisense transcript are rare. In addition to RNA foci and DPR pathology, *C9orf72* patients also develop TDP-43 pathology that correlates well with neurodegeneration like in other forms of FTLD/ALS (Mackenzie *et al*, 2013), but it is still unclear how the *C9orf72* repeat expansion triggers TDP-43 pathology. In contrast, several neuropathology studies failed to detect a strong correlation of the different DPR species (or RNA foci) with the region-specific neurodegeneration seen in *C9orf72* ALS and FTLD patients (Mackenzie *et al*, 2013; Schludi *et al*, 2015), suggesting an interplay of several factors and/or non-cell autonomous effects such as spreading and seeding may be crucial for pathogenesis. Interestingly, $GA_{15}$ peptides form amyloid-like fibrils that are taken up by N2a cells (Chang *et al*, 2016).

Thus, we asked whether poly-GA and the other DPR species are transmitted between cells and how DPR uptake affects the receiving cells. We detected cell-to-cell transmission of all hydrophobic DPR species and show that poly-GA boosts repeat RNA levels and DPR expression, suggesting DPR transmission may trigger a vicious cycle. Treating cells with anti-GA antibodies reduced intracellular aggregation of DPRs. Poly-GA antibodies blocked the seeding activity of *C9orf72* brain extracts which further supports the therapeutic potential of our discovery.

# Results

## Poly-GA and poly-PR differentially affect repeat RNA expression and translation

To allow better interpretation of DPR seeding experiments, we first analyzed DPR protein co-localization in cell lines co-expressing repeat RNA and synthetic DPR constructs. Thus, we cotransfected ATG-initiated synthetic DPR expression plasmids with GFP tag together with a $(G4C2)_{80}$ expression vector driven by the strong CMV promoter (Mori *et al*, 2016). As expected, RAN translation leads to $GA_{80}$-flag aggregation under all conditions. We observed co-aggregation of $GA_{80}$-flag with $GA_{175}$-GFP, but little specific co-localization with the other DPR proteins, which were mainly diffusely localized in the cytoplasm as reported previously (May *et al*, 2014; Zhang *et al*, 2014). Compared to the GFP co-expression, $GA_{80}$-flag aggregates appeared larger particularly in $GA_{175}$-GFP- and $PR_{175}$-GFP-expressing cells and to a lesser extent also with the other DPR proteins (Fig 1A). Quantification confirmed the increased size of $GA_{80}$-flag aggregates in $GA_{175}$-GFP- and $PR_{175}$-GFP-expressing cells and showed no significant effects on the number of aggregates upon co-expression of any DPR species (Fig 1B and C). Similarly, filter-trap analysis showed enhanced aggregation of $GA_{80}$-flag particularly in $GA_{175}$-GFP- and $PR_{175}$-GFP-expressing cells (Fig 1D and E).

Since poly-PR binds RNA and RNA-binding proteins (Kwon *et al*, 2014; Kanekura *et al*, 2016) and thus might affect mRNA

expression, we quantified the expression levels of the repeat mRNA (Fig 1F). Poly-PR had no significant effect on the repeat RNA, suggesting it may mainly induce RAN translation. In contrast, poly-GA expression unexpectedly also increased the levels of the $(G4C2)_n$ RNA.

Together, these data indicate that especially poly-GA and poly-PR proteins promote repeat transcription and/or RAN translation. In contrast to patient tissue, poly-GA did not specifically co-aggregate with the other DPR species under our conditions. Thus, uptake of poly-GA may affect both expression and nucleation in receiver cells.

## Poly-GA, poly-GP, and poly-PA are transmitted between cells

To address whether large DPR proteins are transmitted between cells, we performed co-culture experiments. HEK293 cells were first transfected separately with either DPR-GFP, GFP, or RFP expression vectors. After 24 h, RFP-transfected cells were resuspended and mixed with GFP- or DPR-GFP-transfected cells. Double-positive cells were quantified using flow cytometry analysis immediately after mixing or after 24 h of co-culture (Fig 2A and B). In mixtures of GFP- and RFP-transfected cells, double-positive cells were extremely rare (~0.3%) at both time points. In contrast, $GA_{175}$-GFP was detected in 1–2% of RFP-positive cells after 24 h of co-culture indicating transmission of $GA_{175}$-GFP to RFP-transfected neighboring cells (Fig 2C and D). Furthermore, double-positive cells were sorted to image GFP-tagged DPR proteins in RFP-positive receiver cells (Fig EV1), thus implying secretion and uptake of poly-GA by neighboring cells. We detected even higher intercellular transmission of $GP_{47}$-GFP and $PA_{175}$-GFP, which show mostly diffuse cytoplasmic expression (May *et al*, 2014; Zhang *et al*, 2014).

In contrast, positively charged GFP-$GR_{149}$ and $PR_{175}$-GFP, which localize to cytoplasm and nucleus, were not detected in the RFP-positive receiving cells above background levels. To compensate for the different transfection and expression levels of the GFP-DPR proteins, we also normalized the double-positive cells to the total population of GFP-positive cells (Fig 2D), which showed a similar result compared to the absolute fraction of double-positive cells (Fig 2C). Thus, the hydrophobic cytoplasmic DPR proteins are transmitted between cells regardless of their aggregation properties.

## $GA_{175}$ aggregates seed further poly-GA aggregates in repeat RNA-expressing cells

To test whether transmitted DPR proteins act as a seed for further aggregation, we next used $(G4C2)_{80}$-transfected cells as receiving cells in co-culture experiments. We first confirmed that $(G4C2)_{80}$-transfected cells also take up $GA_{175}$-GFP by co-staining of $GA_{80}$-flag and GA-GFP for analysis by flow cytometry after 3 days of co-culture to allow sufficient levels of RAN translation (Fig 3A and B). We detected a similar fraction of double-positive cells for co-culture of $(G4C2)_{80}$ and $GA_{175}$-GFP-, $GP_{47}$-GFP-, or $PA_{175}$-GFP-expressing cells as with RFP-positive receiver cells (compare Figs 2 and 3B). Since $(G4C2)_{80}$ drives mainly poly-GA expression (Mori *et al*, 2016), we focused on this DPR species for the following experiments.

To further increase the load of transmissible DPR proteins, we incubated $(G4C2)_{80}$-transfected cells for 3 days with $GA_{175}$-RFP aggregates (Fig 3C). Immunofluorescence confirmed intracellular uptake of $GA_{175}$-RFP aggregates (Fig 3D). The exogenous aggregates

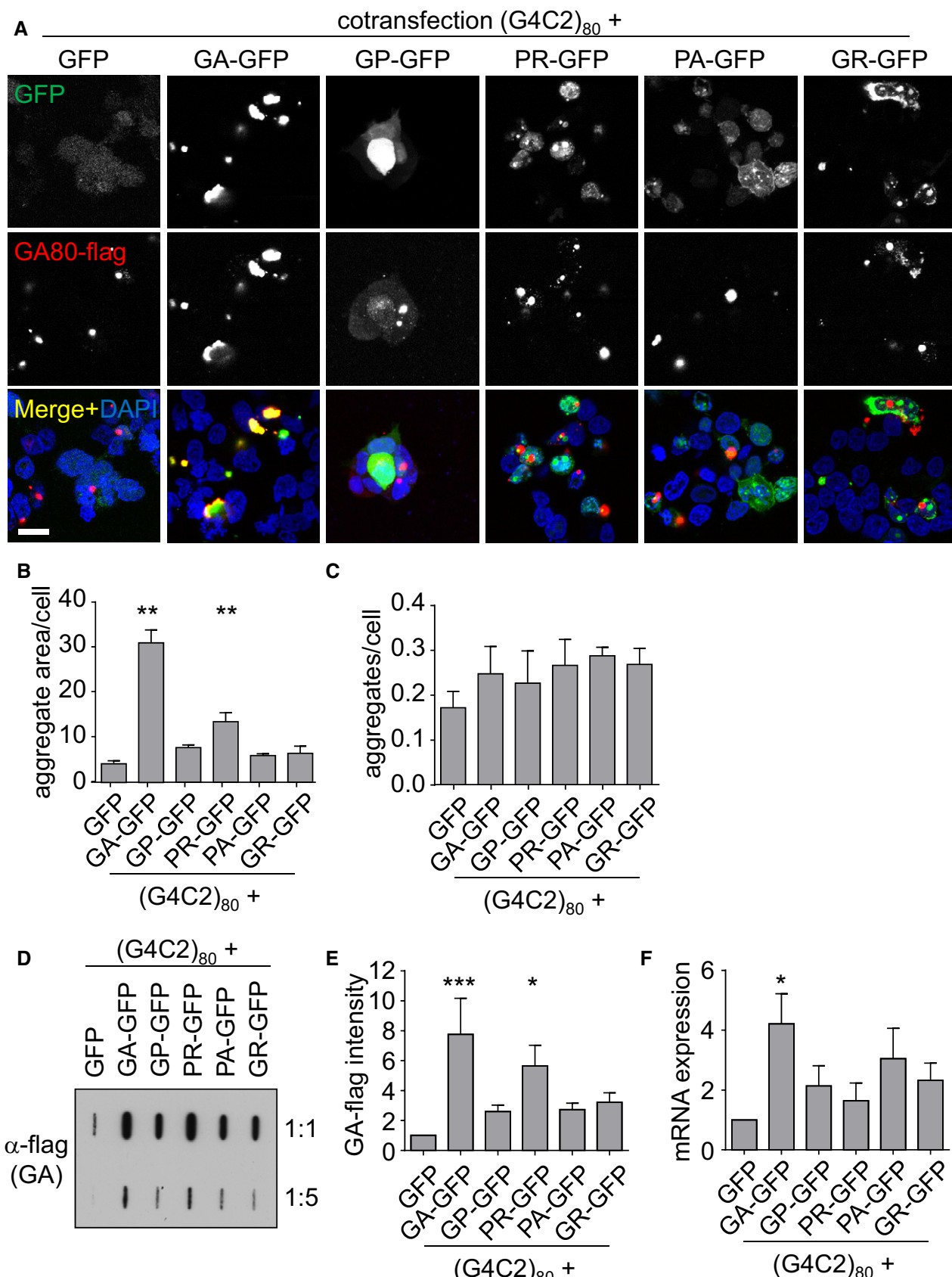

**Figure 1.**

◀

**Figure 1.  DPR expression promotes RAN translation from $(G4C2)_{80}$.**
HEK293 cells cotransfected with $(G4C2)_{80}$ containing a flag-tag in the poly-GA reading frame and GFP or DPR-GFP for 3 days to analyze effects on RAN translation.

A       Immunofluorescence for the GFP-tagged proteins and RAN translation-derived $GA_{80}$-flag. DAPI labels nuclei. Scale bar 20 μm.

B, C    Quantification of $GA_{80}$-flag aggregate area and number from $n = 4$ independent experiments with five images each (containing 60–90 cells per image). Aggregate and cell number were counted manually, and aggregate size was determined by thresholding. Data are shown as mean ± SD. One-way ANOVA with Dunnett's multiple comparisons test; GFP vs. GA-GFP $P = 0.0025$; GFP vs. PR-GFP $P = 0.0095$; **$P < 0.01$.

D       Filter-trap analysis of $GA_{80}$-flag in two dilutions. A representative of four experiments is shown.

E       Quantification of $GA_{80}$-flag from four independent experiments. Data are shown as mean ± SD. Statistics were performed by one-way ANOVA with Dunnett's multiple comparisons test; GFP vs. GA-GFP $P = 0.0009$; GFP vs. PR-GFP $P = 0.0325$; *$P < 0.05$, ***$P < 0.001$.

F       Expression of the $G4C2_{80}$ RNA was measured by qPCR targeting the 3′ region of the repeat sequence. RNA levels were normalized to *GAPDH* mRNA. Data are shown as mean ± SD ($n = 3$). Statistics were performed by one-way ANOVA with Dunnett's multiple comparisons test; GFP vs. GA-GFP $P = 0.0241$; *$P < 0.05$.

co-localized with $GA_{80}$-flag derived from the $(G4C2)_{80}$ vector (Fig 3D, arrow), indicating that transmitted poly-GA can seed further aggregation. Importantly, even cells without prominent $GA_{175}$-RFP staining showed increased $GA_{80}$-flag levels compared to cells treated with RFP extracts, suggesting that even trace amounts of $GA_{175}$-RFP can accelerate poly-GA aggregation in the receiving cells (Fig 3D, arrowhead). Importantly, also the fraction of $GA_{80}$-flag-positive cells increased significantly, suggesting that genuine seeding occurred (Fig 3E).

Filter-trap experiments and flow cytometry analysis confirmed increased expression/aggregation of RAN translation-derived $GA_{80}$-flag and to a lesser extent also of $GR_{80}$-HA and $GP_{80}$-myc in GA-RFP-treated cells on a biochemical level (Fig 3F and G). Similar to direct poly-GA expression (Fig 1F), exposure to $GA_{175}$-RFP lysates also increased the levels of the $(G4C2)_{80}$ mRNA transcripts (Fig 3H), indicating that poly-GA may affect transcription or stability of the expanded *C9orf72* repeat RNA. Taken together, uptake of poly-GA promotes further aggregation of poly-GA, poly-GR, and poly-GP in cells expressing the *C9orf72* repeat expansion.

### Dipeptide repeat proteins promote repeat RNA foci formation

To corroborate the effect of poly-GA on repeat RNA levels, we analyzed nuclear RNA foci, which are another disease hallmark of *C9orf72* FTLD/ALS. We switched from HEK293 to HeLa cells, because they attach better to glass coverslips and can sustain the harsh washing steps for *in situ* hybridization. As $(G4C2)_{80}$ expression resulted in many coalescing RNA foci, which made counting their number unreliable, we analyzed the size of RNA foci. Cotransfection of $GA_{175}$-GFP, $PA_{175}$-GFP, and GFP-$GR_{149}$ significantly increased foci size compared to GFP control, while $GP_{47}$-GFP and $PR_{175}$-GFP expression had no effect (Fig 4A and B). The effects of DPR proteins on RNA foci in HeLa cells are comparable to their effects on repeat RNA levels in HEK293 cells (compare Figs 4B and 1F).

To verify the effects of DPR proteins on the repeat RNA under physiological conditions, we used primary fibroblasts derived from patients with expanded G4C2 repeats and transduced them with individual DPR-GFP-expressing lentiviruses. Since DPR expression in primary patient-derived cells (including induced pluripotent stem cells) is extremely low, we investigated the effect on RNA foci formation. Consistent with the effects of DPR proteins on RNA foci in HeLa cells (Fig 4B), expression of poly-GA, poly-PA, and poly-GR increased the number of foci per cell (Fig 4C and D), whereas poly-PR had no effect on foci formation. Thus, poly-GA, poly-PA, and poly-GR seem to promote transcription or stability of the expanded repeat RNA.

### Poly-GA is transmitted between neurons

To replicate our data in primary neurons, we transduced donor and receiver cells on separate coverslips for 3 days and co-cultured both coverslips with spacers from paraffin dots for another 4 days. We focused on poly-GA and used both $(G4C2)_{80}$ and empty vector-transduced receiver cells. Unfortunately, repeat-transduced neurons show only low $GA_{80}$-flag expression, presumably due to poor packaging efficiency of the repeat RNA (Fig 5A). In contrast, lentiviral transduction of primary neurons with $GA_{175}$-GFP results in inclusions of size and intensity comparable to the aggregates in cortex of *C9orf72* patients (May *et al*, 2014).

Consistent with Figs 2 and 3, we did not detect transmission from the GFP control donor to the receiver cells (Fig 5A, first row). In contrast, we detect $GA_{175}$-GFP inclusions in several receiver neurons after 4 days of co-culture (Fig 5A, second row), suggesting that neurons can release and take up poly-GA similar to HEK293 cells. In addition, we noticed co-localization of transmitted $GA_{175}$-GFP and RAN-translated $GA_{80}$-flag in some receiver cells expressing $(G4C2)_{80}$ (Fig 5A, fourth row).

To directly assess poly-GA release from neurons, we collected conditioned media every 24 h and performed a poly-GA immuno-assay. We first detected poly-GA levels in $GA_{175}$-GFP-transduced cells compared to GFP controls 48 h after transduction (Fig 5B), but poly-GA release was significantly higher on the third and fourth day. Thus, neurons are able to release and take up low levels of poly-GA similar to tau and other intracellular aggregates.

### Brain lysates from *C9orf72* mutation carriers seed poly-GA aggregates in repeat RNA-expressing cells

Next, we asked whether patient-derived DPR aggregates can induce seeding. Therefore, we homogenized cerebella of FTLD patients with or without *C9orf72* mutation, because in this brain region, DPR levels are very high and TDP-43 aggregation is virtually absent (Mackenzie *et al*, 2013). Similar to established protocols for tau seeding, we used liposome-mediated transfection to promote aggregate uptake in $(G4C2)_{80}$-expressing cells (Nonaka *et al*, 2010; Sanders *et al*, 2014).

Cerebellar extracts from *C9orf72* patients increased the number of $GA_{80}$-flag-positive cells compared to *C9orf72*-negative controls as quantified by flow cytometry (Fig 6A and B). Filter trap confirmed the enhanced $GA_{80}$-flag aggregate levels in cells treated with extracts from a *C9orf72* patient compared to a *C9orf72*-negative control (Fig 6C and D). Cerebellar extracts from a *C9orf72* patient also increased the levels of $GR_{80}$-HA and $GP_{80}$-myc (Fig 6C and D).

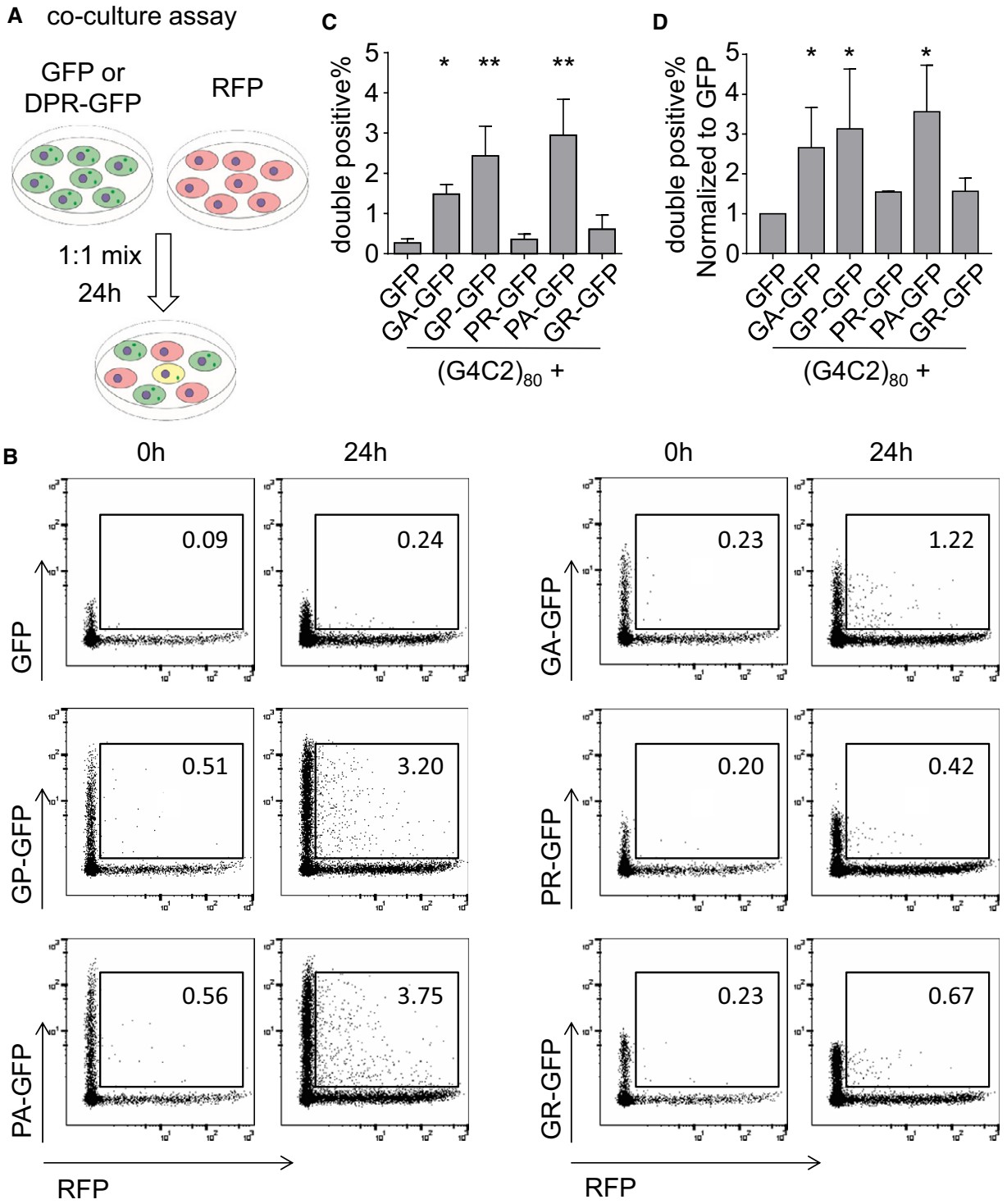

**Figure 2. Hydrophobic DPR proteins are transmitted between cells in co-culture assays.**

HEK293 cells were transfected with RFP, GFP, or DPR-GFP for 24 h and mixed in the indicated combinations. Co-cultures were analyzed by flow cytometry immediately after mixing or 24 h later. Gating was performed on RFP-expressing cells compared to mixture of all green fluorescent cells.

A  Schematic overview of experimental flow.

B  The fraction of double-positive cells is indicated in percent. A representative of three experiments is shown.

C  Absolute frequency of double-positive cells after 24 h of co-culture. Data are shown as mean ± SD ($n = 4$). GFP vs. GA-GFP $P = 0.0482$; GFP vs. GP-GFP $P = 0.0019$; GFP vs. PA-GFP $P = 0.0012$; *$P < 0.05$, **$P < 0.01$ by one-way ANOVA with Dunnett's multiple comparisons test.

D  Relative frequency of double-positive cells to total GFP-expressing cells after 24 h of co-culture. Data are shown as mean ± SD ($n = 4$). GFP vs. GA-GFP $P = 0.0473$; GFP vs. GP-GFP $P = 0.0327$; GFP vs. PA-GFP $P = 0.0166$; *$P < 0.05$ by one-way ANOVA with Dunnett's multiple comparisons test.

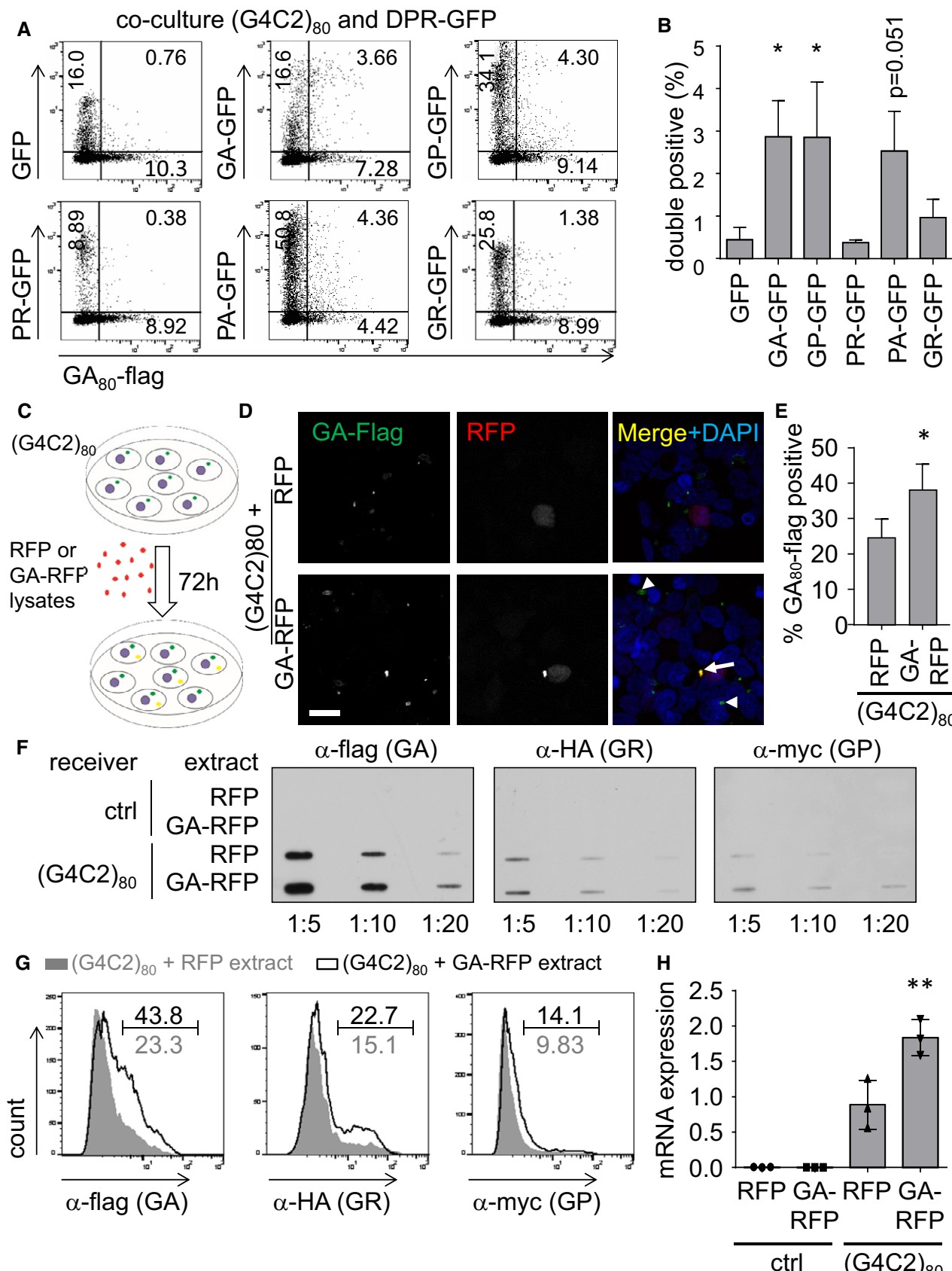

Figure 3.

**Figure 3.  Poly-GA uptake seeds DPR aggregation and induces repeat RNA expression.**

A, B   Co-culture assay in HEK293 cells. 24 h after transfection with either $(G4C2)_{80}$, GFP, or DPR-GFP, cells were mixed in the indicated combination. After 72 h of co-culture, cells were fixed, permeabilized, and stained with anti-flag to detect $GA_{80}$-flag for flow cytometry analysis. Flow cytometry dot plots are shown based on levels of $GA_{80}$-flag (x-axis) and GFP (y-axis) expression. The fraction of indicated populations is indicated in percent. Graphs shows mean ± SD fraction of double-positive cells from three independent experiments. Statistics were performed by one-way ANOVA with Dunnett's multiple comparisons test; GFP vs. GA-GFP $P = 0.0316$; GFP vs. GP-GFP $P = 0.0331$; GFP vs. PA-GFP $P = 0.0513$; *$P < 0.05$.

C–H   HEK293 cells transfected with $(G4C2)_{80}$ for 48 h were treated for 72 h with cell lysates from HEK293 transfected with RFP or $GA_{175}$-RFP as depicted in (C). The RAN-translated $GA_{80}$-flag, $GR_{80}$-HA, $GP_{80}$-myc are detected by anti-flag immunofluorescence (D) and quantified (E). Arrowheads indicate $GA_{80}$-flag aggregates in cells without prominent $GA_{175}$-RFP uptake, arrows indicate co-localization of exogenous $GA_{175}$-RFP with $GA_{80}$-flag. Results from $n = 4$ independent experiments with five images each quantified and analyzed by two-tailed unpaired $t$-test. Data are shown as mean ± SD. $P = 0.0061$; *$P < 0.05$. Scale bar 20 μm. Filter trap (F) and flow cytometry analysis (G) confirmed the increased levels of $GA_{80}$-flag in $GA_{175}$-RFP-treated cells. The percentage of DPR-positive cells in GA-RFP-treated cells compared to the RFP control is indicated. A representative of three independent experiments is shown. (H) Expression of the $(G4C2)_{80}$ RNA in DPR-treated cells was measured by qPCR targeting the tag region downstream of the repeat sequence. RNA levels were normalized to *GAPDH* mRNA. Data are shown as mean ± SD ($n = 3$). Statistics were performed by one-way ANOVA with Dunnett's multiple comparisons test; $(G4C2)_{80}$ + RFP vs. $(G4C2)_{80}$ + GA-RFP $P = 0.007$; **$P < 0.01$.

Similar to the experiments with cell lysates, this was associated with an upregulation of $(G4C2)_{80}$ mRNA expression in the cells receiving extracts from different *C9orf72* mutant patients (Fig 6E). Thus, uptake of patient-derived DPR proteins induces DPR aggregation in (G4C2)-repeat-expressing cells by seeding aggregation and increasing repeat RNA levels.

### Treatment with specific antibodies blocks poly-GA aggregation and seeding

Since antibody treatment has been shown to reduce intracellular aggregation of tau and α-synuclein, which are also known to be transmitted between cells (Boutajangout *et al*, 2011; Chai *et al*, 2011, 2012; Yanamandra *et al*, 2013), we tested whether anti-GA antibodies could inhibit aggregation in our cell culture model. Treating $GA_{175}$-GFP-transfected HEK293 cells with anti-GA reduced $GA_{175}$-GFP aggregation compared to isotype control (Fig 7A and B). Filter-trap assays using a stable cell line expressing $GA_{149}$-GFP confirmed that anti-GA reduced poly-GA aggregate levels compared to isotype control antibodies (Fig 7C). To analyze the efficacy of anti-GA antibodies in neurons, we transduced primary neurons with $GA_{175}$-GFP and treated with antibodies for 6 days (Fig 7D). Treatment with anti-GA significantly reduced poly-GA levels compared to an isotype control (Fig 7E).

We next assessed the ability of anti-GA antibodies to block the seeding activity of brain extracts from *C9orf72* patients on repeat-expressing cells. Brain lysates were pre-incubated with anti-GA or IgG2a control for 16 h and then added to $(G4C2)_{80}$-expressing HEK293 cells for 48 h before measurement. We detected increased expression of $GA_{80}$-flag in cells receiving cerebellar extracts from a *C9orf72* patient (compare Figs 7F and G, and 6A–D). Pre-incubation with anti-GA antibodies reduced the $GA_{80}$-flag expression to control levels, without affecting expression of $GR_{80}$-HA or the repeat RNA levels (Fig EV2), indicating that poly-GA is crucial for the seeding activity of *C9orf72* brains.

Together, these data suggest that anti-GA immunotherapy may prevent seeding and spreading of poly-GA in *C9orf72* disease.

## Discussion

We demonstrate intercellular spreading and seeding of the hydrophobic DPR species poly-GA, poly-GP, and poly-PA. Uptake of poly-GA from transfected cells or from brain homogenates promotes expression of the expanded repeat RNA and RAN translation products, suggesting a vicious cycle of DPR expression and repeat RNA expression. Anti-GA antibodies block the seeding activity of *C9orf72* brain extracts and reduce poly-GA aggregation in cell lines, suggesting immunotherapy may be a useful therapeutic option to treat the DPR component of *C9orf72* disease.

### Hydrophobic DPR proteins are transmitted between cells

Using co-culture assays, we show intercellular transmission of the hydrophobic DPR species poly-GA, poly-GP, and poly-PA in cell lines (Fig 1) and we confirmed poly-GA release and uptake in rat primary neurons (Fig 5). Moreover, cells treated with poly-GA-containing cell extract or *C9orf72* brain homogenates show induced aggregation of RAN-translated $GA_{80}$-flag (Figs 3, 6, and 7).

Our data add to previous reports that fibrillar $GA_{15}$ peptides are taken up by N2a cells and promote intracellular poly-GA aggregation (Chang *et al*, 2016), because we show intercellular transmission of much larger synthetic poly-GA and even patient-derived poly-GA. In contrast to Aβ seeding, which is very inefficient with synthetic peptides and seems to require an elusive cofactor from patient brain (Stohr *et al*, 2012), at least poly-GA seeding seems to work with synthetic peptides and lysates from cell culture or cerebellum. In addition, we detected intercellular spreading of poly-GP and poly-PA. Poly-GP is readily detectable in CSF of *C9orf72* patients (Su *et al*, 2014), but whether extracellular poly-GP in the CSF originates from active secretion or cellular debris is unclear. Our co-culture data rather point to unconventional secretion or passive release of small amounts of hydrophobic DPR proteins as it has been shown for intracellular tau or α-synuclein (Chai *et al*, 2012), because DPR expression is not toxic in HEK293 cells under our conditions (May *et al*, 2014). We did not find significant transmission of arginine-rich DPRs at physiological levels, although synthetic $GR_{20}$ and $PR_{20}$ peptides are taken up by cells and cause toxicity by interfering with RNA expression and splicing when applied at 10 μM (Kwon *et al*, 2014).

While this manuscript was under review, Westergard *et al* reported cell-to-cell transmission of the hydrophobic DPR 50-mers, $GR_{50}$-GFP, and in case of direct cell contact also of $PR_{50}$-GFP (Westergard *et al*, 2016). Even low-level transmission of these species might be relevant due to their high toxicity (Mizielinska *et al*, 2014). The different results between our studies may be due to different repeat length or expression levels, as the arginine-rich

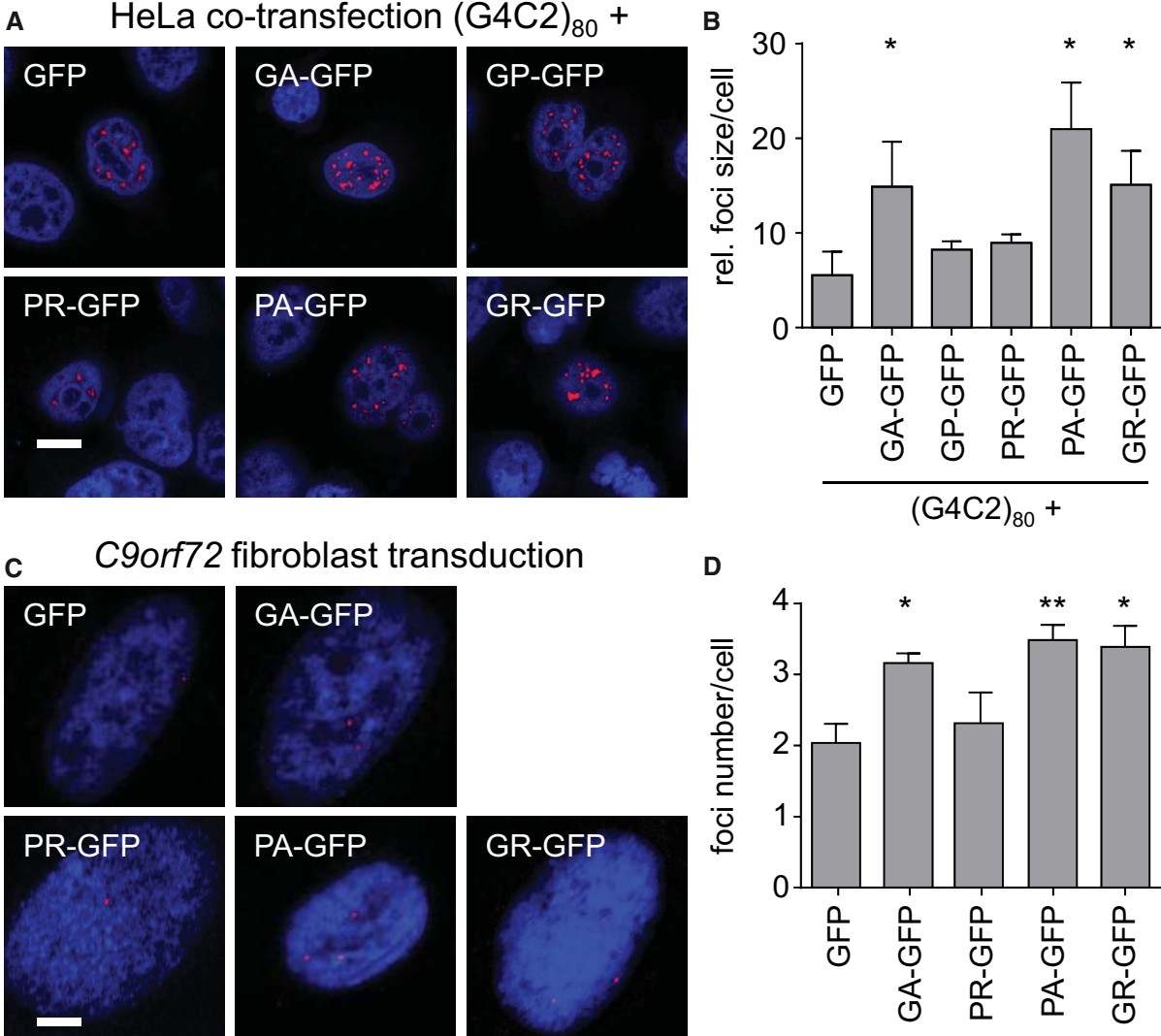

**Figure 4. DPR expression promotes formation of repeat RNA foci in HeLa cells and *C9orf72* fibroblasts.**

A, B  *In situ* hybridization of RNA foci (red) in HeLa cells cotransfected with $(G4C2)_{80}$ and GFP or DPR-GFP for 3 days. Representative images (A) and quantification (B) of foci size from three experiments (at least 30 cells per condition per experiment) are shown. DAPI labels nuclei. Scale bar 10 μm. Summary indicated the means ± SD. GFP vs. GA-GFP $P = 0.0210$; GFP vs. PA-GFP $P = 0.0163$; GFP vs. GR-GFP $P = 0.0413$; *$P < 0.05$ by one-way ANOVA with Dunnett's multiple comparisons test.

C, D  *In situ* hybridization of $(G4C2)_n$ RNA foci in fibroblast of *C9orf72* patients transduced with GFP or DPR-GFP lentivirus for 8–9 days. Note that we could not analyze poly-GP, because we failed to generate a codon-modified lentivirus. Representative images (C) and quantification of foci number (D) are shown. Brightness and contrast were digitally enhanced for better visibility for the presentation only. Scale bar 40 μm. Summary indicated the means ± SEM of $n = 7$ experiments for GFP, GA-GFP, PR-GFP, and PA-GFP, and $n = 3$ for GR-GFP (at least 30 cells per condition per experiment). GFP vs. GA-GFP $P = 0.0296$; GFP vs. PA-GFP $P = 0.0041$; GFP vs. GR-GFP $P = 0.0451$; *$P < 0.05$, **$P < 0.01$ by one-way ANOVA with Dunnett's multiple comparisons test.

DPRs show lower expression in our system (Fig 1 and May *et al*, 2014).

### Dipeptide repeat proteins affect repeat RNA expression and/or stability

Surprisingly, poly-GA uptake did not only promote $GA_{80}$-flag levels, but also increased expression of the other two RAN products poly-GP and poly-GR (Fig 3F and G). These findings complicate interpretation of the data, but two lines of evidence support seeding

of poly-GA. First, poly-GA uptake in recipient cells increased the number of $GA_{80}$-flag inclusions. Second, poly-GA antibody treatment reduced $GA_{80}$-flag aggregation without affecting its mRNA levels.

Moreover, treating cells with poly-GA extracts induced repeat RNA levels (Fig 3H), suggesting an effect on repeat transcription and/or translation. To exclude variable uptake, we transfected DPR expression constructs and analyzed the repeat RNA. In heterologous cells and in patient fibroblasts, poly-GA and poly-PA expression promoted RNA foci formation and poly-GA increased the levels of repeat RNA (Figs 3H and 4). Since neither of the hydrophobic DPR

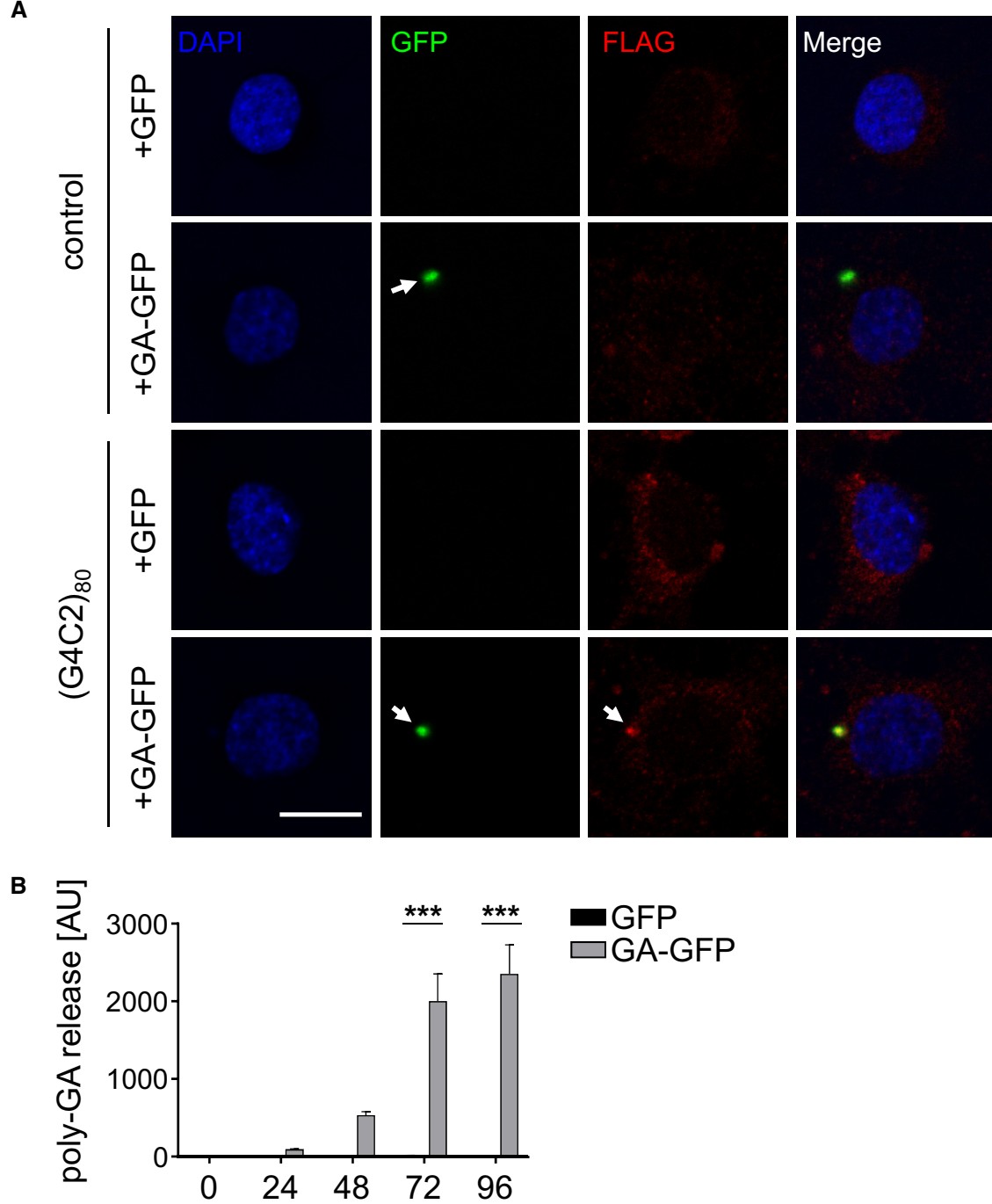

**Figure 5.  Release and uptake of poly-GA by neurons.**

A  Co-culture assay in rat primary neurons. Cortical neurons (400,000/well) on coverslips were transduced with GFP or $GA_{175}$-GFP as donor. Hippocampal neurons (85,000/well) on coverslips were transduced with $(G4C2)_{80}$ or empty vector as receiver cells. Three days later, the washed coverslips were put into well with paraffin spacers. GFP and $GA_{80}$-flag expression was analyzed 4 days later in the receiver cells by immunofluorescence. Arrows indicate co-localization of $GA_{175}$-RFP with $GA_{80}$-flag. Scale bar 10 µm.

B  Cortical neurons transduced with GFP or $GA_{175}$-GFP. Conditioned media were exchanged 24 h prior to transduction and collected right before and every 24 h after infection. Poly-GA levels in media were determined by immunoassay. Data are shown as mean $\pm$ SEM. Two-way ANOVA with Sidak's multiple comparisons test ($n = 4$). $t = 72$ h: GA-GFP vs. GFP ***$P < 0.0001$; $t = 96$ h: GA-GFP vs. GFP ***$P < 0.0001$.

proteins is known to bind RNA or RNA-binding proteins directly, we speculate that the DPR proteins trigger a stress response (Zhang *et al*, 2014) leading to transcriptional upregulation of repeat

transcription. Moreover, about 10% of DPR inclusions are found in the nucleus in patients, where they mainly co-localize with heterochromatin next to the nucleolus, which may support a direct effect

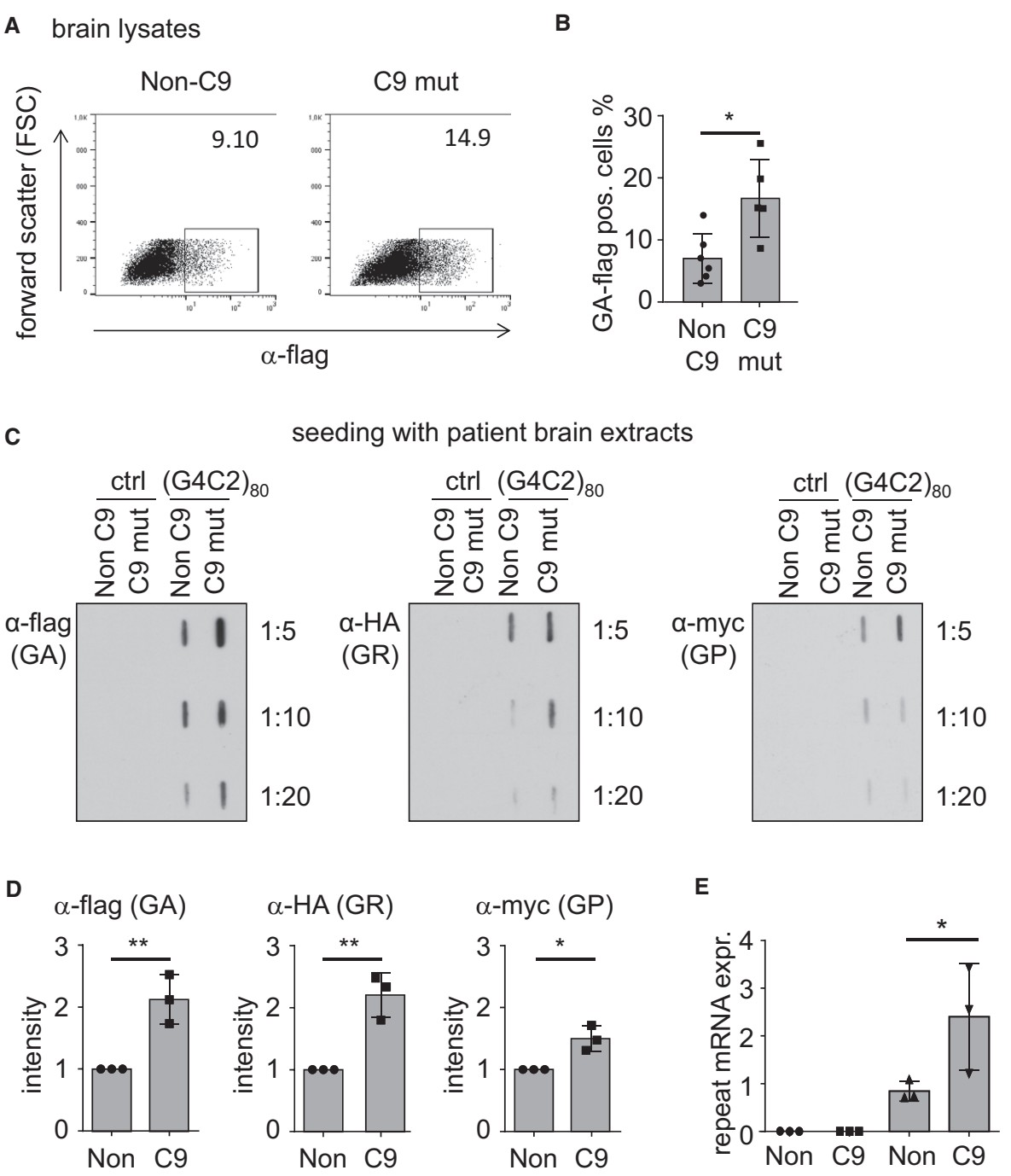

**Figure 6. Brain homogenates from *C9orf72* patients seed DPR aggregation and promote repeat RNA expression.**

Analysis of RAN translation products in HEK293 cells transfected with $(G4C2)_{80}$ (for 24 h) and incubated with cerebellar extracts of *C9orf72* patients and controls.

A, B    Flow cytometry analysis of $GA_{80}$-flag-positive cells using $n = 5$ *C9orf72*-positive and $n = 6$ *C9orf72*-negative cases (three healthy controls, two ALS, one FTLD) $P = 0.0124$; *$P < 0.05$ by two-tailed unpaired $t$-test.

C, D    Filter-trap analysis of DPR products in all three reading frames using the indicated antibodies. Results from $n = 3$ independent experiments using one patient and one control were quantified and analyzed by two-tailed unpaired $t$-test. Data are shown as mean $\pm$ SD. Anti-flag (GA) $P = 0.0079$; anti-HA (GR) $P = 0.0043$; anti-myc (GP) $P = 0.0128$; *$P < 0.05$, **$P < 0.01$.

E    Quantitative RT–PCR shows upregulation of repeat RNA transcripts upon incubation with *C9orf72* extracts as in Fig 3H. Data are shown as mean $\pm$ SD from $n = 3$ patients and controls in independent experiments. Statistics were performed by one-way ANOVA with Dunnett's multiple comparisons test; $(G4C2)_{80}$ + non-C9 vs. $(G4C2)_{80}$ + C9 mut $P = 0.0101$; *$P < 0.05$.

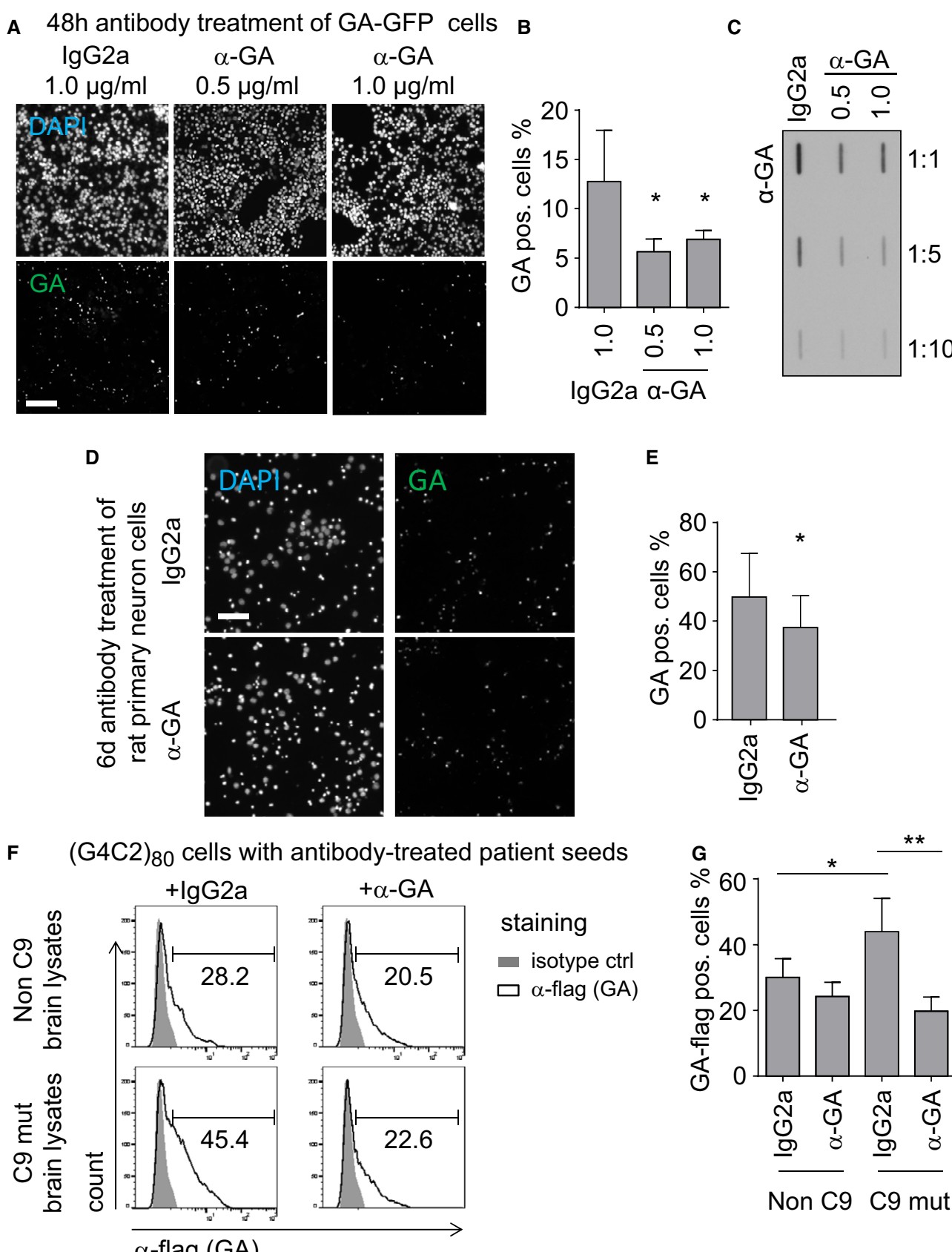

**Figure 7.**

**Figure 7.  Anti-GA antibodies inhibit poly-GA aggregation and prevent seeding from brain tissue.**

A, B  HEK293 cells transfected with $GA_{175}$-GFP were treated with anti-GA antibodies or mouse IgG2a isotype control (in the indicated concentration) for 3 days. Fluorescence microscopy image of GA-GFP aggregation (scale bar 100 μm). (B) The percentage of poly-GA-positive cells was quantified semi-automatically using BioTek Gen5 software. Data are shown as mean ± SD. IgG2a vs. anti-GA 0.5 μg/ml $P = 0.0109$; IgG2a vs. anti-GA 1.0 μg/ml $P = 0.0113$; *$P < 0.05$ by one-way ANOVA with Dunnett's multiple comparisons test from three independent experiments.

C  HEK293-T-REx $GA_{149}$-GFP stable cells cultured in the presence of 10 ng/ml tetracycline were treated with anti-GA antibodies or isotype control as in (A) and analyzed by filter trap. Representative filter-trap blot of three independent experiments is shown.

D, E  Rat primary neurons were transduced with $GA_{175}$-GFP after 5 days *in vitro* (DIV) and treated with 1 μg/ml antibody on the following day. Neurons were analyzed after 6 days of treatment by GFP fluorescence and DAPI staining (scale bar 100 μm). The percentage of poly-GA-positive cells was quantified semi-automatically using BioTek Gen5 software. Data are shown as mean ± SD. $P = 0.0366$; *$P < 0.05$ by two-tailed unpaired *t*-test from $n = 6$ independent experiments.

F, G  HEK293 cells transfected with $(G4C2)_{80}$ were treated with cerebellar extracts pre-incubated with anti-GA or isotype control. The fraction of RAN translation-derived $GA_{80}$-flag was quantified by flow cytometry. Data indicated the means ± SD of $n = 3$ patients and controls in independent experiments. Non-C9 + IgG2a vs. C9 mut + IgG2a $P = 0.0438$; C9 mut + IgG2a vs. C9 mut + anti-GA $P = 0.0013$; *$P < 0.05$, **$P < 0.01$ by one-way ANOVA with Dunnett's multiple comparisons test.

on gene expression (Schludi *et al*, 2015). Surprisingly, poly-PR expression induced poly-GA by RAN translation with little effect on repeat RNA levels or foci formation (Figs 1 and 4). Poly-PR binds directly to RNA and many RNA-binding proteins (Kwon *et al*, 2014; Kanekura *et al*, 2016). Sequestration of certain RNA-binding proteins might impair the tight control of ATG-mediated translational initiation and thus promote RAN translation. Interestingly, antisense oligonucleotides consistently reduce DPR levels stronger than repeat RNA levels independently supporting a feedback mechanism (Jiang *et al*, 2016). Thus, DPR expression may trigger a vicious cycle of increasing repeat RNA and DPR expression ultimately leading to neurodegeneration.

### Poly-GA immunotherapy

Poly-GA, the most abundant DPR protein in patients, could be at the center of *C9orf72* gain-of-function toxicity, because it forms amyloid-like fibrils capable of spreading between cells to seed further DPR aggregation and enhance RNA foci formation. Therefore, we tested whether we could reduce poly-GA aggregation using specific antibodies. Anti-GA antibodies lowered poly-GA levels in both transiently and stably transfected HEK293 cells and also in primary neurons (Fig 7). Moreover, pre-incubation with anti-GA antibodies also prevented uptake from *C9orf72* brain extracts into HEK293 cells (Fig 7).

Immunotherapy targeting extracellular Aβ aggregates has finally shown promising results in patients with Alzheimer's disease in its early stages (Sevigny *et al*, 2016). Surprisingly, anti-tau immunotherapy lowers intracellular tau aggregation and neurological deficits in mouse models (Boutajangout *et al*, 2011; Chai *et al*, 2011, 2012; Yanamandra *et al*, 2013). Even for intracellular aggregates, the antibodies are thought to act on extracellular proteins in transit between two cells. Antibody binding may induce phagocytosis through microglia via Fc receptors or inhibit neuronal uptake (Yanamandra *et al*, 2013). Given our results for cell-to-cell transmission of the different DPR species, only the hydrophobic poly-GA/GP/PA would be accessible for antibodies. Thus, anti-GA immunotherapy may be a future treatment option for *C9orf72* ALS/FTLD. Considering the long prodromal DPR accumulation accompanied by subtle brain atrophy in *C9orf72* patients (Proudfoot *et al*, 2014; Rohrer *et al*, 2015; Edbauer & Haass, 2016), mutation carriers may require very early treatment as proposed for Alzheimer's disease.

Taken together, our work shows an unexpected link between RNA and DPR toxicity and suggests a vicious cycle that may ultimately lead to neuron loss after a prodromal phase. Non-cell autonomous effects due to spreading and seeding of poly-GA, poly-GP,

and poly-PA could explain the poor correlation of DPR proteins and RNA foci with neurodegeneration in *C9orf72* patients and suggest a novel therapeutic approach through passive vaccination.

# Materials and Methods

### Antibodies

The following antibodies were used: anti-DYKDDDDK/flag (filter trap 1:1,000, FACS 1:250, Cell Signaling), anti-myc (1:1,000, clone 9E10, Santa Cruz), anti-HA (1:1,000, clone 3F10, Roche), anti-GFP (1:1,000, clone N86/8, NeuroMab), anti-GA clone 5F2 (1 μg/ml) (Mackenzie *et al*, 2013), mouse IgG2a (1 μg/ml, Sigma), and rabbit IgG (1:250, Sigma).

### Plasmids and lentivirus production

ATG-initiated epitope-tagged synthetic expression constructs for $GA_{175}$-GFP, $PA_{175}$-GFP, GFP-$GR_{149}$, and $PR_{175}$-GFP in pEF6 or lentiviral backbone (FhSynW2) were described previously (May *et al*, 2014; Schludi *et al*, 2015). pEGFP-$GP_{47}$ was a kind gift from Dr. Leonard Petrucelli (Zhang *et al*, 2014) and was for some experiments subcloned into pEF6 vector. The triple-tagged $(G4C2)_{80}$ construct to analyze RAN translation was recently reported (Mori *et al*, 2016). Lentivirus was produced in HEK293FT cells (Life Technologies) as described previously (Fleck *et al*, 2013).

### Cell lines and cell culture

HEK293-T-REx $GA_{149}$-GFP stable cells were generated using T-REx system (Thermo Scientific) according to the manufacturer's instruction. Briefly, $GA_{149}$-GFP was cloned into the pcDNA 5/FRT/TO under the control of CMV promoter and two tetracycline operator 2 (TetO2) sites and transfected in T-REx 293 cells containing the tet-repressor protein. The stable cell line was maintained in high-glucose DMEM medium supplemented with 5 μg/ml blasticidin, 10% FCS, 1% pen/strep, and 2 mM L-glutamine. Expression of $GA_{149}$-GFP was induced with 10 ng/ml tetracycline. HEK293FT cells were cultured with DMEM containing 10% FCS and penicillin/streptomycin.

### Neuron culture

Primary cortical and hippocampal cultures were prepared from E19 rats as described previously (May *et al*, 2014) and plated on

poly-D-lysine-coated coverslips. For co-culture experiments, primary neurons on coverslips with 1 to 2 mm paraffin dots glued on to them were transduced with lentivirus. After 3 days, coverslips were extensively washed and put face to face into one well for 4 days.

### Patient-derived fibroblasts

We included cell lines from three *C9orf72* ALS patients as reported previously (Japtok *et al*, 2015; Mori *et al*, 2016). All procedures were in accordance with the Helsinki convention and approved by the Ethical Committee of the University of Dresden (EK45022009; EK393122012). Patients were genotyped using EDTA blood in the clinical setting after given written consent according to German legislation independent of any scientific study by a diagnostic human genetic laboratory (CEGAT, Tübingen, Germany or Dept. Human Genetics, University of Ulm, Germany) using diagnostic standards.

### Poly-GA immunoassay

Poly-GA in neuronal media was measured by immunoassay on the Meso Scale platform (MSD) using the anti-GA clone 5F2 (Mackenzie *et al*, 2013). Streptavidin plates (MSD Gold 96-well streptavidin) were coated overnight with biotinylated 5F2 antibody (capture antibody, 1:400) in PBS. The next day, the plates were washed three times (0.05% Tween-20, PBS) and blocked for 1 h at room temperature (0.05% Tween-20, 1% BSA in PBS). Plates were incubated with pre-cleared media (5 min, 1,000 *g*) for 2 h at room temperature on a shaking platform. After three washes, the plates were incubated with MSD sulfo-tag-labeled 5F2 antibody (detection antibody, 1:400) for 2 h at room temperature on a shaking platform followed by three final washing steps. Upon adding 100 μl MSD Read Buffer T, the plates were immediately measured. The electrochemical signal was detected using a Meso Scale Discovery SECTOR Imager 2400. After background correction, data are presented in arbitrary units.

### Transfection, immunofluorescence, and filter trap

HEK293FT cells and primary rat neurons were transfected using Lipofectamine 2000 (Thermo Scientific) according to the manufacturer's instructions. For immunofluorescence, cells were fixed with 4% paraformaldehyde and 4% sucrose for 10 min and stained with the indicated antibodies in GDB buffer (0.1% gelatin, 0.3% Triton X-100, 450 mM NaCl, 16 mM sodium phosphate pH 7.4). Images were taken using an LSM710 confocal laser scanning system (Carl Zeiss) with 40× or 63× oil immersion objectives. For filter trap, cells were lysed in Triton buffer (1% Triton X-100, 15 mM $MgCl_2$ in PBS, supplemented with 10 μg/ml DNase and protease inhibitor) on ice. Protein concentration was determined using BCA assay (Thermo Scientific), and equal amount of protein was used. Insoluble pellets were collected by centrifugation at 13,000 rpm/17,949 *g* at 4°C for 30 mins and resuspended in SDS–Tris buffer (2% SDS and 100 mM Tris pH = 7) for 1 h at room temperature. Samples were diluted in SDS–Tris buffer as indicated and filtered through a cellulose acetate membrane (0.2 μm pore).

### Preparation of cell lysates and brain extracts for seeding

Transfected HEK293FT cells or human brain tissue were homogenized in 0.1% Triton X-100 PBS buffer supplemented with DNase, protease inhibitor, and phosphatase inhibitor cocktails, and sonicated for 2 × 20 pulses with 10% amplitude (Branson Digital Sonifier, W-250 D). After brief centrifugation (1,000 *g* for 5 min), the protein concentration in the supernatant was determined using BCA assay (Thermo Scientific). For the seeding assay, 25 μg of cell lysates was applied. To promote aggregate uptake of brain lysates, 25 μg of brain lysates was mixed with 4 μl Lipofectamine 2000 as described previously (Sanders *et al*, 2014). To block the aggregation and spreading of poly-GA, brain lysates were pre-incubated with 2 μg anti-GA antibodies [clone 5F2 (Mackenzie *et al*, 2013)] or mouse IgG2a as control for 16 h.

### Antibody treatment

HEK293 cells transfected with $GA_{175}$-GFP were treated with anti-GA antibodies or mouse IgG2a isotype control at the indicated concentration for 3 days. To assess the efficacy of anti-GA antibodies in neurons, rat primary neurons were transduced with $GA_{175}$-GFP on DIV 5 and treated with anti-GA antibodies or mouse IgG2a isotype control at 1 μg/ml for 6 days. Cells were fixed and counterstained with DAPI. Fluorescence microscopy image of GA-GFP aggregation was taken using Cytation 3 image reader (BioTek). The percentage of poly-GA-positive cells normalized to total cells was quantified semi-automatically using BioTek Gen5 software. For filter trap, HEK293-T-REx $GA_{149}$-GFP stable cells cultured in the presence of 10 ng/ml tetracycline were treated with anti-GA antibodies or isotype control.

### RNA isolation and qPCR

Total RNA was prepared using the RNeasy and QIAshredder kit (Qiagen) according to the manufacturer's instructions. RNA preparations were treated with RNase-Free DNase Set (Qiagen) to minimize residual DNA contamination. 2 μg of RNA was used for reverse transcription with M-MLV Reverse Transcriptase (Promega) using oligo-$(dT)_{12–18}$ primer (Invitrogen). qRT–PCR was performed using CFX384 Real-Time System (Bio-Rad) with TaqMan technology. Primers and probes to the tag region of $(G4C2)_{80}$ construct were designed as described previously (Mori *et al*, 2016). Signals of repeat construct-derived cDNA were normalized to *GAPDH* cDNA according to $\Delta\Delta C_T$ method.

### Flow cytometry and fluorescence-activated cell sorting

HEK293 cells transfected with GFP or RFP were harvested and resuspended in PBS containing 1% FCS and 0.1% (w/v) $NaN_3$ (FACS-PBS). To perform intracellular staining of $GA_{80}$-flag, 1–2 × $10^6$ cells/staining were fixed with 4% PFA for 10 min at 37°C, washed once with PBS, permeabilized with FACS-PBS containing 0.1% (w/v) saponin (FACS-saponin), and incubated with 4% goat serum for 10 min at 4°C to block unspecific binding sites. Cells were then incubated with saturating amount of anti-DYKDDDDK/flag antibody (1:250) or rabbit IgG (1:250) as control for 30 min at 4°C in the dark, followed by a single wash and incubation with saturating amount of secondary antibody (Alexa Fluor 647-labeled anti-rabbit IgG) for 30 min at 4°C. Cells were then washed two times with flow cytometry buffer and analyzed using

**The paper explained**

**Problem**

Expansion of a (G4C2) repeat in *C9orf72* causes FTLD and/or ALS by a gain-of-function mechanism. Patient brains show nuclear foci of the repeat RNA and cytoplasmic aggregates of five DPR proteins that result from non-conventional translation of sense and antisense repeat transcripts in all reading frames (poly-GA, poly-GP, poly-GR, poly-PA, poly-PR). Neither nuclear foci nor DPR inclusions correlate strongly with the areas of neurodegeneration, suggesting non-cell autonomous effects.

**Results**

We show that the hydrophobic DPR proteins poly-GA/GP/PA are transmitted between cells. poly-GA uptake from cell and brain extracts boosts aggregation of all DPR products in receiving cells expressing the repeat RNA. Unexpectedly, poly-GA also promotes repeat RNA expression and foci formation, suggesting a positive feedback loop leading to a vicious cycle of DPR expression and RNA toxicity. Specific antibodies reduce poly-GA aggregation in transfected cells and prevent DPR seeding from patient brain extracts.

**Impact**

Understanding the non-cell autonomous effects of DPR proteins and the positive feedback loop triggering further repeat RNA expression is crucial to elucidate how the global *C9orf72* repeat expansion triggers highly selective neurodegeneration in ALS and FTLD. Blocking this vicious cycle using anti-DPR immunotherapy may help to treat *C9orf72* patients.

MACSQuant VYB (Miltenyi). Data analysis was performed using FlowJo vX software (Treestar).

To perform fluorescence-activated cell sorting of transmitted hydrophobic DPR proteins in a co-culture assay, HEK293 cells were transfected with RFP, GFP, or DPR-GFP for 24 h and mixed in the indicated combination for additional 24 h. Double-positive cells were sorted using a FACSAria Fusion (BD Biosciences) cell sorter and plated on poly-D-lysine-coated coverslips for imaging 17 h later.

### *In situ* hybridization

*In situ* hybridization was performed as described previously with minor changes (Mori *et al*, 2016). Cells were fixed with 4% paraformaldehyde, rinsed twice with SSC, and then incubated in pre-hybridization solution (40% formamide, 2× SSC, 2.5% BSA) at 65°C for 30 min. Cells were then incubated with hybridization solution (40% formamide, 2× SSC, 0.8 mg/ml tRNA (Roche), 0.8 mg/ml single-stranded salmon sperm DNA (Sigma), 0.16% BSA, 8% dextran sulfate (Sigma), 1.6 mM ribonucleoside vanadyl complex (New England Biolabs), 5 mM EDTA, 10 μg/μl 5′ Cy3-labeled 2′-O-methyl-$(CCCCGG)_4$ probe [IDT probe as in (DeJesus-Hernandez *et al*, 2011)] at 65°C for HeLa cells and 60°C for primary human fibroblasts. The following day, cells were sequentially washed with 40% formamide/0.5× SSC for three times 30 min each at 65°C and then with 0.5× SSC three times 10 min each at room temperature. After a brief rinse with PBS, nuclei were counterstained with 0.5 μg/ml of DAPI for 20 min and then washed three times with PBS (3 min each). Glass coverslips were mounted and analyzed on an LSM710 confocal microscope (Carl Zeiss).

### Patient tissue

Patient tissue was collected and provided by the Neurobiobank Munich according to the guidelines of the ethical committee at the Medical Faculty of Ludwig-Maximilians-University (LMU) Munich following the WMA Declaration of Helsinki and the Department of Health and Human Services Belmont Report.

**Expanded View** for this article is available online.

### Acknowledgements

We thank Bettina Schmid, Benjamin Schwenk, and Sabina Tahirovic for critical comments. We thank John Hardy for providing *C9orf72* and control fibroblast cell lines. This work was supported by the NOMIS Foundation (A.H., C.H. and D.E.), Hans und Ilse Breuer Foundation (D.E.), the Munich Cluster of Systems Neurology (SyNergy) (Q.Z., M.S., C.H. and D.E.), Takeda Science Foundation (K.M.) *and* JSPS KAKENHI 16H06953 (K.M.) and the European Community's Health Seventh Framework Programme 617198 [DPR-MODELS] (D.E.). D.B. is supported by the Emmy Noether Programme of the German Research Foundation (DFG; BA 5132/1-1). A.H. and C.H. are supported by the Helmholtz Virtual Institute "RNA dysmetabolism in ALS and FTD (VI-510)".

### Author contributions

QZ and DE conceived the study, analyzed data, and wrote the manuscript with input from all co-authors. QZ, CL, MM, and FS performed experiments. KM, DA, DB, MHS, JG, DF, AF, RF, SM, TA, CK, TK, AH, and CH provided crucial reagents and/or expertise.

### Conflict of interest

CH and DE applied for a patent on DPR detection and immunotherapy.

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
