## [Review Process File · EMBO Molecular Medicine]

Antibodies inhibit transmission and aggregation of C9orf72 poly-GA dipeptide repeat proteins

Qihui Zhou, Carina Lehmer, Meike Michaelsen, Kohji Mori, Dominik Alterauge, Dirk Baumjohann, Martin H. Schludi, Johanna Greiling, Daniel Farny, Andrew Flatley, Regina Feederle, Stephanie May, Franziska Schreiber, Thomas Arzberger, Christoph Kuhm Thomas Klopstock, Andreas Hermann, Christian Haass, and Dieter Edbauer

Corresponding author: Dieter Edbauer, German Center for Neurodegenerative Diseases (DZNE)

Review timeline:	Submission date:	12 September 2016
	Editorial Decision:	19 October 2016
	Revision received:	23 January 2017
	Editorial Decision:	13 February 2017
	Revision received:	23 February 2017
	Accepted:	24 February 2017

Transaction Report:

Editor: Céline Carret

1st Editorial Decision

19 October 2016

Thank you for the submission of your manuscript to EMBO Molecular Medicine and for your recent letter informing us of a similar article just published. We have now heard back from the three referees whom we asked to evaluate your manuscript.

You will see from the comments below that the referees find the manuscript interesting and well performed. Suggestions are provided to make the conclusions stronger and few issues should be clarified and discussed. Referees 2 and 3 recommend repeating the experiments in neurons, which we agree would strengthen the data. You will also see that referees 2 and 3 mention the Cell Rep paper. As you know, this article does not impact our decision at this stage thanks to our scooping protection policy. Nevertheless, in the interest of time, we would strongly encourage you to revise the article to the best of your possibilities but with the understanding that the referees' concerns must be fully addressed and that acceptance of the manuscript would entail a second round of review. We will do our best to facilitate the process our end.

Please note that it is EMBO Molecular Medicine policy to allow only a single round of revision and that, as acceptance or rejection of the manuscript will depend on another round of review, your responses should be as complete as possible.

Revised manuscripts should be submitted within three months of a request for revision; they will otherwise be treated as new submissions, except under exceptional circumstances in which a short

extension is obtained from the editor.

Please have a look at our guidelines and recommendations for submitting your revised article below in order to waste as little time as possible upon submission of the revised article.

I look forward to seeing a revised form of your manuscript as soon as possible.

***** Reviewer's comments *****

Referee #1 (Remarks):

Zhou et al investigate whether the dipeptide repeat proteins (DPRs) generated by the FTD and ALS causing C9orf72 repeat expansion show cell to cell transmission and seeded aggregation. This is an important topic which is of great interest to the field and beyond.

Overall it is clearly demonstrated that HEK and HeLa cells can take up poly-GA protein and that this drives increases in repeat RNA and GA protein levels, which can be reversed by pre-incubation with anti-GA antibodies. As both RNA and protein levels are increased it is not possible to determine whether increased protein levels are due to increased seeded aggregation or simply increased mRNA, leading to increased protein levels. Therefore, the current conclusion that seeded aggregation explains the results is not justified.

Points to address

1. The authors must distinguish between increased poly-GA-induced GA-flag levels/aggregates being caused by increased repeat RNA levels or seeded aggregation.
2. Figure 3c. It is essential to quantify the number of aggregates. For genuine seeding, increased flag aggregates would be expected in GA-RFP treated cells compared to RFP alone.
3. Figure 6. Does the antibody treatment reduce the brain homogenate-induced increase in repeat RNA levels? This will help clarify whether the reduction in poly-GA protein is due to reduced seeded aggregation or reducing the repeat RNA.
4. It should be stated whether the filter trap assays were normalised for protein amount prior to loading, or if a loading control was performed on the blot. If not, the rationale for this should be explained.
5. Figure 2. It would be helpful to see images to observe the nature of the GFP and RFP double positive cells - are there double-labelled aggregates?
6. Figure 2. The conditions which show higher amounts of GFP in RFP cells (GA, GP and PA) also show higher amounts of GFP-only cells. This could indicate that increased transfection efficiency of these constructs is driving the observed effect. The authors should address whether it is more appropriate to normalise to GFP levels.
7. Figure 1. Quantification of aggregate size and number is needed to confirm the effect of DPR co-transfection on (G4C2)₈₀-GA-FLAG.
8. Stating in the title that the antibodies are 'immunotherapeutic' is an unnecessary stretch - immunoprecipitating GA protein from brain lysates does not give a good indication of likely therapeutic potential.

If these points can be addressed, the findings would be an important step forward for the field.

Referee #2 (Comments on Novelty/Model System):

The authors used different cell lines throughout the manuscript. They used HEK293 cell line for

most of their experiment and they used HeLa cell line for some of their assays. The authors could use a more consistent approach or provide a justification for using multiple cell lines.

They should consider using a neuronal cell line or iPSC-derived motor neurons from C9 patients to support their findings. Human fibroblast cells from C9 patients are not the best model for addressing cell-to-cell transmission.

Referee #2 (Remarks):

The manuscript by Zhou and colleagues describe that the hydrophobic DPR proteins are transmitted between cells and expression of poly-GA induces the formation of nuclear RNA foci in (G4C2)₈₀ expressing cells and patient fibroblasts. Hexanucleotide repeat expansion in C9orf72 is the most common cause of ALS/FTD and non-ATG mediated translation (RAN) has been reported in ALS/FTD patient carrying expanded hexanucleotide repeats as well as cellular and animal models of C9orf72. Interestingly, they observed that treatment with recombinant poly-GA and cerebellar extracts of C9orf72 patients elevated repeat RNA levels and promoted aggregation of all DPR proteins in recipient cells expressing (G4C2)₈₀. They went on to demonstrate that treatment with anti-GA antibodies prevented intracellular poly-GA aggregation and blocked the seeding process. Based on these findings, the authors suggest that poly-GA based immunotherapy may help in suppressing disease progression and aggregation in C9orf72 ALS/FTD patients. Overall, the manuscript is well-written and experiments are properly described. There are few concerns and comments that should be addressed.

1. Figure 1A: The authors should describe how many cells were observed. What is the rationale for co-transfecting (G4C2)₈₀ along with GA, GP, PR, PA or GR? Did you use (G4C2) with lower repeats as a control?
2. PR and GR have been shown to be the most toxic DPRs in different cellular and animal models. Why PR and GR are not transmitted between cells? Is it possible that PR and GR proteins are expressed at very low level and that is why it is difficult to observe any cell-to-cell transmission?
3. It is not clear why did they use HEK293 for most of their assays and HeLa cell line for some of their experiments. The authors should use a neuronal cell line that might be more appropriate and relevant to ALS/FTD.
4. Figure 4C and 4D: What about poly-GR? Why poly GR was excluded in this experiment?
5. The authors should consider adding few more controls or C9-ALS patient brain homogenates in the figure 5. Using just one ALS patient sample and one control is not enough for drawing any conclusions.
6. Recently, Westergard et al., (Cell Reports 2016) reported cell-to-cell transmission of DPRs using cell culture models and suggest the all of the DPRs are transmitted cell-to-cell. The authors should discuss this paper in their results and discussion section.
7. There is few syntax and grammatical errors in the manuscript that should corrected.

Referee #3 (Comments on Novelty/Model System):

Non-neuronal cells are used throughout the manuscript. Ultimately, the cell to cell transmission, the induction of repeat mRNA and the immunotherapy should be confirmed in neurons.

Referee #3 (Remarks):

The manuscript by Zhou et al, describes the cell to cell transmission of aggregates formed by RAN translation of dipeptides from C9Orf72 repeat expansions.

The data is clearly interesting, and the manuscript describe the novel findings well. However, cell to cell transmission of C9Orf72 dipeptides has been very recently described by others (Chang et al, JBC, 2016). Also, it has been just published in the context of neurons and patient-derived iPSCs derived motor neurons (Westergard et al, Molecular Cell, 2016).

The novel findings from this manuscript include:

- The identification of a positive feedback loop by which the polyGA dipeptide or cerebellar extracts from C9orf72 patients promote the C9orf72 repeat mRNA expression and/or stability in non-neuronal cells.

- Immunotherapy against the polyGA peptide reduces GA intracellular aggregation and rescues the effect of the polyGA on repeat mRNA levels.

The data presented is comprehensive, but there are a number of issues that would need to be addressed prior publication, including:

1. All experiments have been done in non-neuronal cells. Is this because in the models used, the levels of expression achieved are toxic to primary neurons and/or neuronal cell lines? Ultimately, the cell-to-cell transmission, the induction of repeat mRNA and the immunotherapy should be confirmed in neurons.
2. The cellular models generally used in here rely on the ectopic expression of labelled DPRs and their transmission to "recipient" cells expressing (G4C2)₈₀ constructs. This may lead to artefacts due to over-expression of the "recipient" and "donor" constructs in terms of secretion as well as in terms of the effects on aggregation. The authors addressed this issue by using patient-derived fibroblasts transduced by GFP-DPRs, as "DPR expression in primary patient derived cells is extremely low, they instead focus on RNA foci formation". They indeed found that RNA foci in the "recipient" fibroblasts were increased by the expression of the hydrophobic DPR species by the "donor" cells. However, again the system relies on ectopic expression of DPRs. To avoid any potential artefact due to ectopic expression, the authors should look at transmission between cells expressing physiological levels of the repeats. For example, did they look at the potential effect of patient brain extracts on patient-derived fibroblasts, even at extended time points? As the positive feedback loop should be activated in this system, it's possible that DPR aggregation may be visible in fibroblasts.
3. In *Drosophila* C9Orf72 models, both arginine DPRs (poly-GR and poly-PR) have been shown to be the most toxic DPR species (Mizielinska et al, Science, 2014, not referenced in the manuscript). In the cellular systems tested here, both poly-GR and -PR are the only two DPRs that do not seem to be transmitted between cells. However, transmitted polyGA are shown to increase the aggregation of poly-PR in "recipient" cells, despite no co-localization of GA and PR aggregates in this model. For the immunotherapy experiments, the authors tested the anti PolyGA antibody, but only against "recipient" cells expressing poly-GA. Particularly on the experiments with the cerebellar extracts from patients, it may be possible that the reduction via immunotherapy on the polyGA being transmitted between cells may have an effect on the aggregation of other DPRs.

Minor points:

1. For the RNA foci experiments, HeLa cells are used instead of the HEK293 cells mainly used throughout the manuscript. Why is that? Is it because RNA foci in HEK293 cells were not be reliably counted?
2. Have the authors tried with conditioned media from cultures of "donor cells" instead of mixing "donor and recipient" cultures? Do they seed aggregation and/or repeat RNA as well?
3. TDP-43 co-localized with DPRs in C9Orf72 patients (at least outside the cerebellum). Have the authors look into TDP-43 pathology in their cellular models?

Response to the reviewers

We provide new data to strengthen our main conclusion that poly-GA is transmitted between cells and uptake can be inhibited by anti-GA antibodies. Most importantly, we confirmed our key findings in primary neurons and performed the suggested experiments to support the seeding component of our effects.

- We confirmed cell to cell transmission of poly-GA in primary neurons in a co-culture assay (new Fig. 5A).
- In neurons, we also show release of poly-GA into the media using a novel ELISA (new Fig. 5B).
- Anti-GA antibodies inhibit aggregation in poly-GA expressing primary neurons (new Fig. 7D/E).
- We show that treatment with poly-GA extracts increases the number of poly-GA inclusions in receiving cells, suggesting that uptake of poly-GA leads to nucleation of new inclusions (new Fig. 3E).
- We increased the number of patients and controls for the treatment study with brain extracts in Figure 6. We now show enhanced poly-GA translation in (GGGGCC)₈₀ expressing cells treated with extracts from *C9orf72* patients (n=5) compared to controls (n=6) in the new Fig. 6A/B.
- For the transmission experiments in HEK293 cell we now show images of double positive cells (new Fig. EV1).

In addition, we provide the requested controls experiments (new Fig. 1B/C and EV2). Large text changes are labeled in red in the manuscript.

***** *Reviewer's comments* *****

Referee #1 (Remarks):

Zhou et al investigate whether the dipeptide repeat proteins (DPRs) generated by the FTD and ALS causing C9orf72 repeat expansion show cell to cell transmission and seeded aggregation. This is an important topic which is of great interest to the field and beyond.

Overall it is clearly demonstrated that HEK and HeLa cells can take up poly-GA protein and that this drives increases in repeat RNA and GA protein levels, which can be reversed by pre-incubation with anti-GA antibodies. As both RNA and protein levels are increased it is not possible to determine whether increased protein levels are due to increased seeded aggregation or simply increased mRNA, leading to increased protein levels. Therefore, the current conclusion that seeded aggregation explains the results is not justified.

We thank the reviewer for the overall enthusiasm. We provide additional experimental evidence for genuine seeding and discuss our results more critically in the revised manuscript. While exciting, the dual effect on repeat RNA and DPRs does complicate the interpretation of the results. We assume that both seeding and RNA induction occur in DPR treated cells.

Points to address

1. The authors must distinguish between increased poly-GA-induced GA-flag levels/aggregates being caused by increased repeat RNA levels or seeded aggregation.

We followed the reviewer's helpful suggestions to address whether genuine seeding occurred. Poly-GA donor cells induce the number of GA₈₀-flag inclusions in the receiver cells suggesting that nucleation by uptake of extracellular aggregates occurs (new Fig. 3E). Antibody treatment reduced intracellular poly-GA aggregation without affecting mRNA levels suggesting that preventing the nucleation is most important for reducing DPR aggregation (new Fig. S2B). From our data it seems most likely that both seeding and DPR-mediated repeat RNA induction contribute to the observed effects. We modified our discussion accordingly.

In addition, Westergard et al. have reported that poly-GA/GP/GR/PA are transmitted between cells in several paradigms while our manuscript was under review. However, they focused on receiver

cells not expressing repeat RNA or DPR proteins. Together, these data strongly support cell-to-cell transmission of DPR proteins. Especially uptake of the highly aggregation prone poly-GA is likely to trap DPRs translated in receiver cells.

2. Figure 3c. It is essential to quantify the number of aggregates. For genuine seeding, increased flag aggregates would be expected in GA-RFP treated cells compared to RFP alone.

We followed this excellent suggestion and quantified the fraction of GA₈₀-flag positive cells. GA₁₇₅-RFP treatment indeed increased the number of GA₈₀-flag positive cells, suggesting genuine seeding occurred. The new data is shown in the new Fig. 3E of the revised manuscript.

3. Figure 6. Does the antibody treatment reduce the brain homogenate-induced increase in repeat RNA levels? This will help clarify whether the reduction in poly-GA protein is due to reduced seeded aggregation or reducing the repeat RNA

We performed qPCR analysis as requested and now show that anti-GA antibodies have no effect on repeat RNA levels, arguing that the seeding component is dominant in this experiment (new data in Fig. S2B).

4. It should be stated whether the filter trap assays were normalised for protein amount prior to loading, or if a loading control was performed on the blot. If not, the rationale for this should be explained.

For all filter traps we quantified protein levels from the soluble fraction using BCA assay. This important information is now included in the revised method section.

5. Figure 2. It would be helpful to see images to observe the nature of the GFP and RFP double positive cells - are there double-labelled aggregates?

We now provide images from double positive cells sorted by flow cytometry confirming the transmission of hydrophobic DPR proteins into RFP expressing receiver cells (new Fig. EV1).

6. Figure 2. The conditions which show higher amounts of GFP in RFP cells (GA, GP and PA) also show higher amounts of GFP-only cells. This could indicate that increased transfection efficiency of these constructs is driving the observed effect. The authors should address whether it is more appropriate to normalise to GFP levels.

We reanalyzed our data and normalize it to the GFP positive cells. The different normalization did not affect the overall result. We also tried to manipulate the expression levels using different promoters (CMV, synapsin), which unfortunately affected the expression levels of poly-GA only slightly (data not shown). We discuss the potential confound of our analysis in the revised manuscript.

7. Figure 1. Quantification of aggregate size and number is needed to confirm the effect of DPR co-transfection on (G4C2)₈₀-GA-FLAG.

We quantified the inclusion size and numbers as requested. The new data is shown in the new Fig 1B/C and confirmed the impression from the sample images previously shown in Fig 1A.

8. Stating in the title that the antibodies are 'immunotherapeutic' is an unnecessary stretch - immunoprecipitating GA protein from brain lysates does not give a good indication of likely therapeutic potential.

We agree that the original title may have been overly enthusiastic. We rephrased the title into "Antibodies inhibit transmission and aggregation of C9orf72 poly-GA dipeptide repeat proteins". If these points can be addressed, the findings would be an important step forward for the field.

Referee #2 (Comments on Novelty/Model System):

The authors used different cell lines throughout the manuscript. They used HEK293 cell line for most of their experiment and they used HeLa cell line for some of their assays. The authors could use a more consistent approach or provide a justification for using multiple cell lines.

We provide a justification for switching from HEK293 to HeLa cells in the revised manuscript. Briefly, HeLa cells are better for foci staining for two reasons. HeLa cells have larger nuclei that are easy to observe. Most importantly, HeLa cells attach better to the glass cover slip and can sustain the harsh washing steps during *in situ* hybridization much better than HEK293 cells.

They should consider using a neuronal cell line or iPSC-derived motor neurons from C9 patients to support their findings. Human fibroblast cells from C9 patients are not the best model for addressing cell-to-cell transmission.

While expression of DPR proteins in iPSC derived neurons have been reported by others, several collaborators could not detect DPR proteins using our highly specific monoclonal antibodies. Thus, DPR proteins are likely only visible under highly optimized conditions in iPSC derived neurons. Recently, we could finally detect DPR proteins in iPSC-derived neurons after 3 months of maturation. However, this system is not suitable for a revision time frame. Instead, we now show several new experiments with primary rat neurons confirming transmission of poly-GA (new Fig. 5A) and inhibition of poly-GA aggregation using antibodies (new Fig. 7D/E). Moreover, we detected release of poly-GA from neurons using a novel ELISA (new Fig. 5B).

Referee #2 (Remarks):

The manuscript by Zhou and colleagues describe that the hydrophobic DPR proteins are transmitted between cells and expression of poly-GA induces the formation of nuclear RNA foci in (G4C2)80 expressing cells and patient fibroblasts. Hexanucleotide repeat expansion in C9orf72 is the most common cause of ALS/FTD and non-ATG mediated translation (RAN) has been reported in ALS/FTD patient carrying expanded hexanucleotide repeats as well as cellular and animal models of C9orf72. Interestingly, they observed that treatment with recombinant poly-GA and cerebellar extracts of C9orf72 patients elevated repeat RNA levels and promoted aggregation of all DPR proteins in recipient cells expressing (G4C2)80. They went on to demonstrate that treatment with anti-GA antibodies prevented intracellular poly-GA aggregation and blocked the seeding process. Based on these findings, the authors suggest that poly-GA based immunotherapy may help in suppressing disease progression and aggregation in C9orf72 ALS/FTD patients. Overall, the manuscript is well-written and experiments are properly described. There are few concerns and comments that should be addressed.

We thank the reviewer for the constructive feedback and now address all points in the revised manuscript.

1. Figure 1A: The authors should describe how many cells were observed. What is the rationale for co-transfecting (G4C2)80 along with GA, GP, PR, PA or GR? Did you use (G4C2) with lower repeats as a control?

We provide quantification and the requested statistical information for Fig. 1A in the new Fig. 1B/C and the revised figure legend. We also try to explain the rational in results better. Essentially, we performed these experiments as control after finding effects of DPR expression on repeat-RNA levels.

While it is an interesting idea, we could not analyze effects in low repeat constructs, because they do not support RAN translation (Mori et al., Science 2013).

2. PR and GR have been shown to be the most toxic DPRs in different cellular and animal models. Why PR and GR are not transmitted between cells? Is it possible that PR and GR proteins are expressed at very low level and that is why it is difficult to observe any cell-to-cell transmission?

This is a valid concern. In the meantime Westergard et al. (Cell Rep 2016) have reported transmission of poly-GR and in some conditions also of poly-PR. Kwon et al. (Science 2014) had already shown uptake of short GR and PR peptides. In contrast, we could not detect transmission of poly-GR/PR in our system. We cannot exclude this is due to different expression levels or repeat length and mention this in the revised discussion.

3. It is not clear why did they use HEK293 for most of their assays and HeLa cell line for some of their experiments. The authors should use a neuronal cell line that might be more appropriate and relevant to ALS/FTD.

We used HeLa cells for foci staining purely for technical reasons. HeLa cells have larger nuclei that are easier to observe. Moreover, HeLa cells attach much better to the glass cover slip than HEK293 cells and can sustain harsh washing steps during *in situ* hybridization. We now mention this in the result section.

As requested, we now repeated our key findings in rat primary neurons. We detected secretion of poly-GA by ELISA (new Fig. 5B) and show transmission of poly-GA in a co-culture assay (new Fig. 5A). Moreover, anti-GA antibodies reduce poly-GA aggregation in primary neurons (Fig. 7D/E)

4. Figure 4C and 4D: What about poly-GR? Why poly GR was excluded in this experiment?

Poly-GR was omitted in the original submission, because we did not have enough GFP-GR149 virus for the first experiments. Now we repeated the experiment now three more times with all constructs including poly-GR and show the combined data in the revised Fig. 4C/D.

During the revision experiments we realized that the poly-GP expressing virus contained the original GGGGCC-repeat RNA, which explains the large effect on RNA foci shown in the original submission. We therefore had to remove that data from the revised manuscript, because we have not succeeded to subclone a full length GFP-GP47 cDNA into a viral backbone despite major efforts, as this sequence is extremely unstable in bacteria. The omission is explained in the revised figure legend. We would like apologize for our mistake.

5. The authors should consider adding few more controls or C9-ALS patient brain homogenates in the figure 5. Using just one ALS patient sample and one control is not enough for drawing any conclusions.

We followed this is very reasonable suggestion. We now performed flow cytometry analysis of GA₈₀-flag levels in cells treated with cerebellar extracts from *C9orf72* patients (n=5) or *C9orf72* negative controls (n=6). This data is shown in the new Fig. 6A/B. The mRNA levels from Fig 6E (originally 5C) had been performed from 3 patients and controls, which we now clearly mention in the revised legend.

6. Recently, Westergard et al., (Cell Reports 2016) reported cell-to-cell transmission of DPRs using cell culture models and suggest the all of the DPRs are transmitted cell-to-cell. The authors should discuss this paper in their results and discussion section.

Indeed, Westergard et al. show transmission of GA₅₀/GP₅₀/GR₅₀/PA₅₀-GFP and under some conditions also of PR₅₀-GFP. We could only detect transmission of the hydrophobic GA₁₇₅/GP₄₇/PA₁₇₅-GFP. The difference could be due to different expression levels and repeat length (149 vs 50 repeats for poly-GR). Uptake of PR₂₀/GR₂₀ has been shown Kwon et al. (Science 2014). We now discuss the similarities and discrepancy in the revised manuscript.

7. There is few syntax and grammatical errors in the manuscript that should corrected.

We did our best to fix all of these mistakes in the revised manuscript.

Referee #3 (Comments on Novelty/Model System):

Non-neuronal cells are used throughout the manuscript. Ultimately, the cell to cell transmission, the induction of repeat mRNA and the immunotherapy should be confirmed in neurons.

This is a reasonable request. For the revised manuscript, we replicated our key findings in primary neurons. A major difficulty was the low RAN translation level from neurons transduced with a repeat RNA lentivirus, which we attribute to poor packaging efficacy. This also precluded a reliable analysis of the repeat RNA levels. As we could barely detect poly-GA but not the other DPRs in transduced neurons, we focused on transmission of poly-GA. For details see below.

Referee #3 (Remarks):

The manuscript by Zhou et al, describes the cell to cell transmission of aggregates formed by RAN translation of dipeptides from C9Orf72 repeat expansions.

The data is clearly interesting, and the manuscript describe the novel findings well. However, cell to cell transmission of C9Orf72 dipeptides has been very recently described by others (Chang et al, JBC, 2016). Also, it has been just published in the context of neurons and patient-derived iPSCs derived motor neurons (Westergard et al, Molecular Cell, 2016).

The novel findings from this manuscript include:

- *The identification of a positive feedback loop by which the polyGA dipeptide or cerebellar extracts from C9orf72 patients promote the C9orf72 repeat mRNA expression and/or stability in non-neuronal cells.*

- *Immunotherapy against the polyGA peptide reduces GA intracellular aggregation and rescues the effect of the polyGA on repeat mRNA levels.*

The data presented is comprehensive, but there are a number of issues that would need to be addressed prior publication, including:

We thank the reviewer for the positive evaluation of our work and followed the helpful suggestions in the revised manuscript.

1. All experiments have been done in non-neuronal cells. Is this because in the models used, the levels of expression achieved are toxic to primary neurons and/or neuronal cell lines? Ultimately, the cell to cell transmission, the induction of repeat mRNA and the immunotherapy should be confirmed in neurons.

This is a very reasonable request. We now confirm transmission of poly-GA between neurons growing on coverslips separated by paraffin spacers to prevent direct cell contact (new Fig. 5A). Moreover, we quantified extracellular release of poly-GA from neurons using a novel ELISA (new Fig. 5B). Finally, we include novel data that show therapeutic effect of anti-GA antibodies in neurons expressing poly-GA from synthetic constructs (new Fig 7D/E). Thus, we could replicate our key findings in rat primary neurons.

2. The cellular models generally used in here rely on the ectopic expression of labelled DPRs and their transmission to "recipient" cells expressing (G4C2)₈₀ constructs. This may lead to artefacts due to over-expression of the "recipient" and "donor" constructs in terms of secretion as well as in terms of the effects on aggregation. The authors addressed this issue by using patient-derived fibroblasts transduced by GFP-DPRs, as "DPR expression in primary patient derived cells is extremely low, they instead focus on RNA foci formation". They indeed found that RNA foci in the "recipient" fibroblasts were increased by the expression of the hydrophobic DPR species by the "donor" cells. However, again the system relies on ectopic expression of DPRs. To avoid any potential artefact due to ectopic expression, the authors should look at transmission between cells expressing physiological levels of the repeats. For example, did they look at the potential effect of patient brain extracts on patient-derived fibroblasts, even at extended time points? As the positive feedback loop should be activated in this system, it's possible that DPR aggregation may be visible in fibroblasts.

We agree with the reviewer that exaggerated expression levels might affect transmission. However, transmission between donor and receiver with endogenous levels will be extremely difficult to show, because endogenous donor and receiver DPR proteins cannot be distinguished without sophisticated labeling strategies.

We have shown previously that the inclusions in poly-GA transduced neurons are comparable to patient tissue in both size and intensity (May et al., Acta Neuropath 2014). Under these conditions we could detect release and uptake of poly-GA in a co-culture assay (new Fig. 5A). Moreover, we had already analyzed the repeat RNA induction at endogenous RNA levels using patient fibroblasts by foci staining (Fig 4C/D). Moreover, we have used endogenous DPR proteins from the patient brain extracts as donor for the antibody therapy (Fig. 7).

3. In Drosophila C9orf72 models, both arginine DPRs (poly-GR and poly-PR) have been shown to be the most toxic DPR species (Mizielinska et al, Science, 2014, not referenced in the manuscript). In the cellular systems tested here, both poly-GR and -PR are the only two DPRs that do not seem to be transmitted between cells. However, transmitted polyGA are shown to increase the aggregation of poly-PR in "recipient" cells, despite no co-localization of GA and PR aggregates in this model. For the immunotherapy experiments, the authors tested the anti PolyGA antibody, but only against "recipient" cells expressing poly-GA. Particularly on the experiments with the cerebellar extracts from patients, it may be possible that the reduction via immunotherapy on the polyGA being transmitted between cells may have an effect on the aggregation of other DPRs.

We followed the reviewer's excellent suggestion and analyzed RAN-translation derived poly-GR in repeat expressing cells treated with brain extracts (new Fig. EV2). *C9orf72* brain extracts induced poly-GR levels and repeat mRNA, but the anti-GA antibodies reduced poly-GA expression without affecting poly-GR or RNA levels. We assume that the other DPR proteins not targeted by our anti-GA antibody may elicit this response.

Minor points:

1. For the RNA foci experiments, HeLa cells are used instead of the HEK293 cells mainly used throughout the manuscript. Why is that? Is it because RNA foci in HEK293 cells were not be reliably counted?

We used HeLa cells for foci staining purely for technical reasons. HeLa cells have larger nuclei that are easier to observe. Moreover, HeLa cells attach much better to the glass cover slip than HEK293 cells and can sustain harsh washing steps during *in situ* hybridization. We now mention this in the result section.

2. Have the authors tried with conditioned media from cultures of "donor cells" instead of mixing "donor and recipient" cultures? Do they seed aggregation and/or repeat RNA as well?

We have now shown poly-GA release from primary neurons and have used a co-culture experiment without direct cell contact using two separately transduced coverslips separated by paraffin spacers (new Fig. 5A). In addition, we we have analyzed extracellular release of poly-GA in the new Fig. 5B.

3. TDP-43 co-localized with DPRs in C9orf72 patients (at least outside the cerebellum). Have the authors look into TDP-43 pathology in their cellular models?

TDP-43 and DPRs indeed colocalize in a small subset of inclusions in patients. However, we have not observed this so far in our cellular models. However, we recently reported that poly-GA overexpression induces cytoplasmic TDP-43 granules that do not colocalize with poly-GA (Khosravi et al., HMG in press). Thus, uptake of poly-GA may have additional effects on TDP-43.

2nd Editorial Decision

13 February 2017

Thank you for the submission of your revised manuscript to EMBO Molecular Medicine. We have now received the enclosed reports from the referees that were asked to re-assess it. As you will see the reviewers are now supportive and I am pleased to inform you that we will be able to accept your manuscript pending the following final amendments:

1) please indicate in legends exact n= and exact p= values, not a range. Some people found that to keep the figures clear, providing a supplemental table with all exact p-values was preferable. You

are welcome to do this if you want to.

Please submit your revised manuscript within two weeks. I look forward to seeing a revised form of your manuscript as soon as possible.

***** Reviewer's comments *****

Referee #1 (Remarks):

The authors have addressed my concerns. I have no further comments.

Referee #2 (Remarks):

NA

Referee #3 (Remarks):

The major revisions done by the authors make the manuscript acceptable for publication

2nd Revision - authors' response

23 February 2017

Thank you very much for the positive news. We made all changes as requested. Only for Fig. 4 we cannot provide the exact p-value, because GraphPad Prism only states $p < 0.0001$ for very low p values. We didn't want to change the statistical test, although a 2000-fold difference would be significant in any test. I hope you can accept this.

Corresponding Author Name: Dieter Edbauer

Manuscript Number: EMM-2016-07054